# When Additive Noise Meets Unobserved Mediators: Bivariate Denoising Diffusion for Causal Discovery

**Dominik Meier**[*]
Cornell Tech
dm954@cornell.edu

**Sujai Hiremath**[*]
Cornell Tech
sh2583@cornell.edu

**Promit Ghosal**
University of Chicago
promit@uchicago.edu

**Kyra Gan**
Cornell Tech
kyragan@cornell.edu

## Abstract

Distinguishing cause and effect from bivariate observational data is a foundational problem in many disciplines, but challenging without additional assumptions. *Additive noise models* (ANMs) are widely used to enable sample-efficient bivariate causal discovery. However, conventional ANM-based methods fail when unobserved mediators corrupt the causal relationship between variables. This paper makes three key contributions: first, we rigorously characterize why standard ANM approaches break down in the presence of unmeasured mediators. Second, we demonstrate that prior solutions for hidden mediation are brittle in finite sample settings, limiting their practical utility. To address these gaps, we propose *Bivariate Denoising Diffusion* (BiDD) for causal discovery, a method designed to handle latent noise introduced by unmeasured mediators. Unlike prior methods that infer directionality through mean squared error loss comparisons, our approach introduces a novel independence test statistic: during the noising and denoising processes for each variable, we condition on the other variable as input and evaluate the independence of the predicted noise relative to this input. We prove asymptotic consistency of BiDD under the ANM, and conjecture that it performs well under hidden mediation. Experiments on synthetic and real-world data demonstrate consistent performance, outperforming existing methods in mediator-corrupted settings while maintaining strong performance in mediator-free settings.

## 1 Introduction

Determining the causal direction between two variables (X→Y) is fundamental to scientific domains ranging from genomics to economics. However, traditional discovery methods, such as constraint-based [61, 60] and scoring-based methods [8, 27, 17] can only identify causal graphs up to an equivalence class, leaving them unable to distinguish the causal direction between a variable pair. Additional assumptions are necessary to enable bivariate discovery [46], and they mostly fall under three categories: (1) *the location scale noise model* (LSNMs), (2) *the principle of independent mechanisms* (ICM), and (3) the additive noise model (ANM).

LSNMs express the outcome $Y$ with heteroskedastic, multiplicative noise relative to the treatment $X$, i.e. $Y = f(X) + g(X)\varepsilon$, where $\varepsilon \perp\!\!\!\perp X$. While LSNMs allow for increased flexibility, existing approaches require additional parametric assumptions for identifiability [62, 64, 18, 7]. ICM approaches assume that the marginal distribution of the cause and the conditional mechanism generating the effect

---

[*]Shared first co-author.

39th Conference on Neural Information Processing Systems (NeurIPS 2025).

are independent components of the *data-generating process* (DGP) [57, 22]. While they impose no explicit functional form, these methods rely on unverifiable structural asymmetries [39, 19], often fail under non-invertible mechanisms [20], and often lack theoretical guarantees [62]. In contrast, *additive noise models* offer unique advantages for bivariate discovery, allowing for consistent recovery of causal directions without strong parametric assumptions [67], permitting sample complexity characterization under Gaussian noise [70], and enabling polynomial-time guarantees for global discovery on large graphs [47]. These properties have spurred both methodological advances [35, 13, 65, 14] and real-world applications [54, 28].

However, these strengths vanish when hidden variables corrupt the observed causal relationships—a near-ubiquitous scenario in real-world systems like biomedicine [28] and economics [2]. Indeed, as Peters et al. [48] point out, although the joint distribution of all variables may admit an ANM, the joint distribution over a subset that excludes some mediators may not allow for an ANM (see Appendix D.3). To the best of our knowledge, despite the rapid advances in statistical tests that handle unobserved confounding of causal pairs [21, 33, 34, 66, 30], only one bivariate discovery method [6] addresses the problem of unobserved mediators. However, Cai et al. [6] provides no correctness guarantees, requires nonlinearity, and has poor empirical performance (Section 5). This leaves a glaring gap in practical bivariate causal discovery.

**Contributions.** In this paper, we propose *bivariate denoising diffusion* (BiDD), a causal direction identification method that works for general ANM, even in the presence of unobserved mediators. Our contributions are fourfold:

- **Analysis of Unmeasured Mediators:** We first introduce the ANM-UM, a novel approach for modeling unobserved mediators (Section 2). We then characterize how unobserved mediators break the ANM assumption over observed variables, finding that this occurs if and only if there are nonlinear mechanisms induced after the initial transformation of the cause (Lemma 2.3).

- **Failure-Mode of Existing Methods:** We first categorize conventional ANM-based methods into three types: Residual-Independence, MSE-Minimization, and Score-Matching based (Section 3). For each category, we show that existing methods will fail to correctly recover the directionality when unobserved mediators break the ANM assumption (Lemmas 3.1-3.4). We then analyze the only method developed to handle hidden mediation, discussing potential issues.

- **Diffusion Methodology and Guarantees** We develop BiDD, a practical alternative to existing ANM based methods, hypothesizing that the noise predictions from a conditional diffusion model will be less dependent on the condition when the condition is the cause, rather than the effect (Section 4). We show a consistency result under the assumption of an ANM (Theorem 4.2), and conjecture that a similar result may hold in the ANM-UM setting.

- **Comprehensive Evaluation:** We extensively evaluate BiDD on synthetic data, demonstrating that only our approach is able to achieve uniformly strong performance across DGPs with linear, nonlinear non-invertible, and nonlinear invertible mechanisms (Section 5.2). We then validate BiDD on a large real world dataset, the Tübingen Cause-Effect pairs [41], where it achieves comparable results to the best baselines, highlighting BiDD's robustness across diverse domains (Section 5.2). The source code for BiDD is publicly available.[2]

## 2  Problem Setup

In this work, we focus on the discovery of the causal direction between a causal pair $(X, Y)$, which is generated by an *ANM with Unobserved Mediators* (ANM-UM). In this section, we first formally introduce the structural causal model describing ANM-UM. We then establish identifiability conditions and characterize when ANM-UM cannot be simplified to standard ANMs. A complete notation table is included in Appendix A.

Under ANM-UM, the outcome Y is generated from cause X through unobserved mediators $\{Z_i\}$ (Figure 1), with each $Z_i$ introducing independent noise while remaining unmeasured. Formally, given $T$ unmeasured mediators, the DGP between $X$ and $Y$ can be described as follows:

---

[2]https://github.com/dommeier/bidd

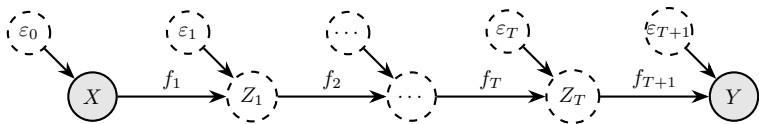

Figure 1: ANM-UM (Eq 2.1), where mediators $Z_1, \ldots Z_T$ and noises $\varepsilon_0, \ldots \varepsilon_{T+1}$ are all unobserved.

$$\begin{cases} Z_1 = f_1(X) + \varepsilon_1, \\ Z_2 = f_2(Z_1) + \varepsilon_2, \\ \vdots \\ Z_T = f_T(\text{Pa}(Z_T)) + \varepsilon_T, \\ Y = f_{T+1}(\text{Pa}(Y)) + \varepsilon_{T+1}, \end{cases} \tag{2.1}$$

where $X, \{\varepsilon_i\}_{i \in [T]}$ are mutually independent. The functions $\{f_1, \ldots, f_{T+1}\}$ can be linear or nonlinear, and the $\varepsilon$ can be arbitrary (Gaussian or non-Gaussian). We choose to model the process with multiple latent variables, instead of collapsing all $Z_i$ into one $Z$, to preserve the independent additive noise structure.

**Assumption 2.1** (ANM-UM Setting). *Suppose $X, Y$ follow ANM-UM described by Eq. (2.1). Then, we assume: 1) no confounders among $X, \{Z_i\}_{i \in [T]}$, and $Y$; 2) acyclicity; 3) no selection bias (noise independence is preserved in the data collection process); 4) $X \not\perp\!\!\!\perp Y$, i.e., $f$ is nonzero almost everywhere (otherwise, $X \perp\!\!\!\perp Y$, detectable via simple independence testing).*

The conventional ANM and the related Post-Nonlinear (PNL) Model [67] are special cases of ANM-UM. ANM corresponds to zero mediators (i.e., $T = 0$), while PNL corresponds to one mediator (i.e., $T = 1$) and no additive noise on $Y$ (i.e., $\varepsilon_{T+1} = 0$). Our ANM-UM also generalizes the Cascade Additive Noise Model (CANM) [6], which assumes all functions $\{f_1, \ldots, f_{T+1}\}$ are nonlinear. See Appendix D.1 for details.

Prior work (Theorem 1, Cai et al. [6]) shows that certain joint distributions over $(X, Y)$ admit ANM-UM representations in both directions $X \dashrightarrow Y$ and $Y \dashrightarrow X$, rendering the causal direction unidentifiable without further assumptions. We refer to such distributions as *Backward ANM-UM*, as the structural assumptions of ANM-UM hold in both the causal and anticausal directions. These distributions can only arise under pathological conditions, such as when all functions are linear and the noises are Gaussian. We thus impose:

**Assumption 2.2** (Identifiability Constraint). *Under Eq. (2.1) and Assump. 2.1, no backward ANM-UM exists where $X = g(Y, \hat{\varepsilon}_1, \ldots, \hat{\varepsilon}_T) + \hat{\varepsilon}_{T+1}$ with $Y, \hat{\varepsilon}_1, \ldots, \hat{\varepsilon}_{T+1}$ mutually independent.*

Appendix D.2 provides explicit constraints on the backward mechanism $g$ and noise terms $\{\hat{\varepsilon}_i\}_{i \in [T]}$ that preclude non-identifiability under Assumption 2.2.

While CANM [6] requires all mediators to be nonlinear, ANM-UM permits identifiability with unobserved mediators even under linear transformations, reducing to standard ANM when the causal effect admits an additive decomposition:

$$Y = A_1(X) + A_2(\varepsilon_1, \ldots, \varepsilon_T) + \varepsilon_{T+1}, \tag{2.2}$$

where functions $A_1$ and $A_2$ (determined by $f_1, \ldots, f_T$) are separable without $X$-$\varepsilon$ interaction terms. For example, in a 3 variable ANM-UM $X \to Z_1 \to Y$, if $f_1$ is nonlinear and $f_2$ is linear, it reduces to the ANM setting, whereas if $f_1, f_2$ are both nonlinear, it does not (see Appendix D.3). Lemma 2.3 (proof in Appendix D.4) formalizes this: ANM-UM is irreducible to ANM if and only if there exists a mediator $Z_i$ such that $Y$ depends nonlinearly on $Z_i$:

**Lemma 2.3** (Irreducible ANM-UM and Nonlinear Mediator). *Under ANM-UM (Eq. (2.1)) and Assump.s 2.1 and 2.2, $Y$ does not admit a decomposition in Eq. (2.2) if and only if there exists a mediator $Z_i$ such that $Y = h(Z_i) + \tilde{\varepsilon}$ for some nonlinear $h$, with $\tilde{\varepsilon} \perp\!\!\!\perp Z_i$. Additionally, we call such a mediator $Z_i$ a* nonlinear mediator.

# 3 Failure-Modes of Prior Work

In this section, we illustrate how both standard ANM methods and one existing hidden mediator approach fail under ANM-UM settings (Eq.(2.1) and Assumption 2.1), assuming identifiability (Assumption 2.2) and irreducibility of ANM-UM (Lemma 2.3).

## 3.1 Traditional ANM-based Bivariate Methods

Existing methods mostly fall into three categories: 1) *Residual-Independence* (RI): identify the cause via an independent residual, 2) *Score-Matching*: identify the effect via conditions on the score function, 3) *MSE-Minimization*: identify the cause via the smallest residual. For each class, we present its core decision rule and construct ANM-UM counterexamples where it fails.

**Residual-Independence** Key methods include DirectLiNGAM [59], its nonparametric generalization RESIT [47], and PNL [67]. The former two leverage ANM-induced residual independence asymmetries via a common decision rule (Decision Rule E.2): if the residual from regressing Y on X is independent of X but the residual from regressing X on Y depends on Y, we conclude X causes Y, and vice versa. If both residuals are independent or dependent, no conclusion can be drawn.

As a counterexample, consider the following ANM-UM: $X \sim \mathcal{N}(0,1), Z = X^2 + \varepsilon_1, Y = Z^2 + \varepsilon_2$, with $\varepsilon_1, \varepsilon_2 \sim \mathcal{N}(0,1)$. The residual $e_1 := Y - \mathbb{E}[Y|X]$ can be simplified as

$$e_1 = Y - \mathbb{E}[Y|X] = Y - \mathbb{E}[(X^4 + \varepsilon_1^2 + \varepsilon_2) + (2\varepsilon_1 X^2)] = \varepsilon_1^2 + \varepsilon_2 + 2\varepsilon_1 X^2 - 1. \quad (3.1)$$

Since $e_1 \not\perp\!\!\!\perp X$, Decision Rule E.2 does not return the correct causal directionality and fails to identify $X \dashrightarrow Y$. We formalize this intuition in Lemma 3.1 (proof in Appendix E.3):

**Lemma 3.1** (Regression Residual-Independence Fails). *Assuming a consistent estimator for regression residuals and access to infinite data, Decision Rule E.2 fails to identify the correct causal direction when at least one mediator is nonlinear.*

PNL assumes a more complicated structure between $X, Y$: $Y = f(g(X) + \varepsilon_1)$, where $g, \varepsilon_1, f$ are the nonlinear effect, independent noise, and invertible post-nonlinear distortion, respectively. As $\varepsilon_1$ can be represented as the difference $f^{-1}(Y) - g(X)$, [67] proposes to identify the causal direction by recovering independent noise. If they can find functions $l_1, l_2$ such that $e_1 \perp\!\!\!\perp X$ for $e_1 = l_2(Y) - l_1(X)$, then they say that the causal hypothesis $X \dashrightarrow Y$ 'holds' (Decision Rule E.3).

While valid for restricted ANM-UM cases (e.g., single nonlinear mediator), this approach may fail with multiple mediators due to $f$'s invertibility requirement (Lemma 3.2, proof in Appendix E.5).

**Lemma 3.2** (PNL Residual-Independence Fails). *Assuming a consistent ICA residual estimator and access to infinite data, Decision Rule E.3 fails to recover the correct causal direction when there exists at least one non-invertible nonlinear mediator.*

Prior work [59, 47, 67] propose alternative rules to compare measures of dependence, rather than independence, to improve finite sample performance (see Appendix E.4 for more details). However, empirically, we find this heuristic often fails (see Section 5).

**Score-Matching** The original score-matching method SCORE [52] (with several followup works [36, 56] leveraging the same fact) relies on the assumption of Gaussian noise and nonlinear mechanisms to identify the effect via a condition on the Jacobian of the score function ($\nabla \log p(X, Y)$). Montagna et al. [35] prove that SCORE can fail to correctly decide causal direction when the noise is non-Gaussian, proposing NoGAM as a noise agnostic solution for nonlinear ANM. They further extend NoGAM to Adascore [37], which they prove correctly recovers the causal direction for all identifiable ANM.

Adascore identifies the causal direction by proving that only the residual from nonparametrically regressing the effect onto the cause is a consistent estimator of a particular expression involving the score (Rule E.4). However, their theory relies on the estimated residual being independent from the cause, which, as demonstrated in Eq. (3.1) may be false in some ANM-UM. Thus, Decision Rule E.4 fails to identify $X \dashrightarrow Y$. We formalize this intuition in Lemma 3.3 (proof in Appendix E.6)

**Lemma 3.3** (Score-Matching Fails). *Assuming a consistent estimator of the conditional expectation and access to infinite data, Decision Rule E.4 fails to recover the correct causal direction when there exists at least one nonlinear mediator.*

**MSE Minimization** Key methods include CAM [5], NoTEARS [68] and GOLEM [43], and NoTEARS-MLP [69]. The causal direction is determined by comparing prediction error: whichever variable better predicts the other (lower MSE) is designated the cause (Rule E.5). While effective in some synthetic settings, this rule suffers from two key flaws: 1) standardizing degrades performance [50], and 2) the $L_2$ loss is only lower in the causal direction under restrictive variance conditions [44], which may not hold under ANM-UM (Lemma 3.4, proof in Appendix E.7). Intuitively, the causal direction becomes unidentifiable when $R^2$-sortability vanishes (i.e., equal prediction errors in both directions), a problematic limitation since DGPs may exhibit arbitrary $R^2$ values [51].

**Lemma 3.4** (MSE-Minimization Fails). *Given a consistent estimator of the conditional expectation and infinite data, Rule E.5 fails to recover the correct causal direction when $\mathbb{E}[Var[X|Y]] < \mathbb{E}[Var[Y|X]]$.*

## 3.2 Hidden Mediator Method—CANM

Assuming nonlinear mechanisms, CANM [6] uses a variational autoencoder (VAE) framework to: 1) learn latent noise via VAE ($X, Y \rightarrow \mathcal{N}(\mu, \sigma^2)$), 2) compare the *evidence lower bound* (ELBO, Eq. (3.2)) scores for both causal directions, and 3) infer causation via higher ELBO (Rule E.6). Critically, CANM minimizes a reconstruction loss combined with a regularization term on the latent space, which does not align directly with the assumptions of the ANM-UM. This means that, while CANM succeeds empirically on synthetic non-invertible Gaussian DGPs, it lacks theoretical guarantees, even for the standard ANM without mediators. We further note that VAE-based methods in general require specifying the dimensionality of the latent space, i.e., the number of unobserved mediators, which is difficult to estimate and can affect performance. In contrast, denoising diffusion does not require estimating this unknown parameter. Therefore, for tasks where aspects of the underlying causal structure (such as the number of mediators) are both important and hard to ascertain, we propose that it is theoretically justified to prefer diffusion over VAEs.

Our experiments (Section 5) show failure cases with: 1) linear/invertible mechanisms, 2) non-Gaussian noise (often exhibiting posterior collapse, see Appendix E.9). As VAE training often encounters posterior collapse in practice, next we examine the behavior of CANM under this phenomenon. Posterior collapse causes CANM's learned $\mathcal{N}(\mu, \sigma^2)$ to degenerate to $\mathcal{N}(0, 1)$. This eliminates the KL term in ELBO and reduces the objective to the sum of negative entropy of $X$ and the conditional log-likelihood of $Y|X$ (Eq. (3.3)):

$$\text{ELBO}_{X \rightarrow Y} = \mathbb{E}[\log p(x)] + -\beta \text{KL}\left(q_\phi(n \mid x, y) \,\|\, p(n)\right) + \mathbb{E}_{n \sim q_\phi(n|x,y)} \log p\left(\varepsilon = y - f(x, n; \theta)\right) \tag{3.2}$$

$$= -H(X) + E \log p_{Y|X}\left(\varepsilon = y - f(x; \theta)\right). \tag{3.3}$$

When posterior collapse occurs, CANM is provably inconsistent for ANM-UM where this sum is not higher for the causal direction (Lemma 3.5, proof in Appendix E.10):

**Lemma 3.5** (CANM Fails). *Assuming infinite data and a consistent estimator of the conditional expectation, Rule E.6 fails to recover the causal direction if posterior collapse occurs and the expected conditional log-likelihood minus the entropy is higher in the causal direction.*

## 4 Bivariate Causal Discovery Using Diffusion

In this section, we develop our conditional diffusion-based method for distinguishing between cause and effect generated by the ANM-UM. We first warm up by developing intuition in the linear setting about when denoising leads to predicted noise that is independent of one of its input variables. We then spell out a decision rule for deciding the causal direction that leverages the developed intuition, providing theoretical guarantees of correctness under certain restrictions of the ANM-UM. We end by introducing a practical method for denoising-diffusion for bivariate discovery, BiDD, and providing its computational complexity.

### 4.1 Denoising and Independence

To better understand what asymmetries may arise from denoising in the causal vs. anticausal direction, we start with a simplified setup, restricting ANM-UM to only linear mechanisms without unobserved

mediators. We let the DGP of $X, Y$ follow

$$Y = X + \varepsilon_1, \ X \perp\!\!\!\perp \varepsilon_1,$$

where $\varepsilon_1$ is non-Gaussian (to ensure identifiability [59]). To show the asymmetry in this setup, we formulate two noising processes: one where keep $X$ fixed and noise $Y$, and one where we keep $Y$ fixed and noise $X$. In the noising processes, we inject independent Gaussian noise into the noised variable, obtaining the noised terms

$$\widetilde{X} = X + \varepsilon_X \ \ \text{and} \ \ \widetilde{Y} = Y + \varepsilon_Y, \ \ \varepsilon_X, \varepsilon_Y, \sim \mathcal{N}(0, 1). \tag{4.1}$$

Now, in the denoising processes, we aim to find the best estimators $f_{\varepsilon_X}, f_{\varepsilon_Y}$ such that MSE losses

$$(\varepsilon_Y - f_{\varepsilon_Y}(\widetilde{Y}, X))^2 \ \ \text{and} \ \ (\varepsilon_X - f_{\varepsilon_X}(\widetilde{X}, Y))^2 \tag{4.2}$$

are minimized. Intuitively, the unnoised variable contains information about the noised one, so including it can enhance noise prediction and reduce the loss. However, this inclusion may also introduce dependence between the predicted noise and the unnoised variable. Consider the case where $X$ is the cause and $Y$ is the effect. Under the ANM-UM assumption (Assumption 2.1), this implies that $Y$ is a function of $X$ and independent noise terms. Let $(X_i, Y_i)$ be a fixed sample from the dataset. Suppose we add noise to $Y_i$ to obtain a noised version $\tilde{Y}_i = Y_i + \varepsilon_{Y_i}$. When predicting $\varepsilon_{Y_i}$ from $\tilde{Y}_i$ and $X_i$, the estimate $\varepsilon_{\hat{Y}_i}$ may depend on $X_i$, since $X_i$ contains information about the true value of $Y_i$. This induces statistical dependence between $X$ and the predicted noise $\varepsilon_{\hat{Y}_i}$. The same argument applies in the anticausal direction. However, the dependence is not symmetric, as the functional relationship between $X$ and $Y$ is not symmetric. This asymmetry in dependence may provide a useful signal for causal identification and motivates our use of a conditional diffusion model. Specifically, we expect that the independence test outcomes for the pairs $(X, f_{\varepsilon_Y}(\widetilde{Y}, X))$ and $(Y, f_{\varepsilon_X}(\widetilde{X}, Y))$ to differ. We now formalize this intuition.

**Causal Direction: Denoising $\widetilde{Y}$ and Testing Independence between $X$ and $f_{\varepsilon_Y}(\widetilde{Y}, X)$** Given infinite data, the best estimators $f_{\varepsilon_Y}^*, f_{\varepsilon_X}^*$ of the MSE loss converges to the conditional expectation [35]. This implies that the prediction of injected $\varepsilon_Y$ equals

$$\hat{\varepsilon}_Y = \mathbb{E}[\varepsilon_Y \mid \widetilde{Y}, X]. \tag{4.3}$$

Substituting $Y = X + \varepsilon_1$ into $\widetilde{Y} = Y + \varepsilon_Y$, we have:

$$\widetilde{Y} - X = \varepsilon_1 + \varepsilon_Y. \tag{4.4}$$

Next, we will show that $\widetilde{Y} - X$ is a sufficient statistics for $\mathbb{E}[\varepsilon_Y \mid \widetilde{Y}, X]$, i.e., $\mathbb{E}[\varepsilon_Y \mid \widetilde{Y}, X] = \mathbb{E}[\varepsilon_Y \mid \widetilde{Y} - X]$. To see this, we observe that since $X \perp\!\!\!\perp \varepsilon_1$ and $X \perp\!\!\!\perp \varepsilon_Y$, we have $\varepsilon_Y \perp\!\!\!\perp X \mid \varepsilon_1 + \varepsilon_Y \implies \varepsilon_Y \perp\!\!\!\perp X \mid \widetilde{Y} - X$. This implies that

$$\hat{\varepsilon}_Y = \mathbb{E}[\varepsilon_Y \mid \widetilde{Y}, X] = \mathbb{E}[\varepsilon_Y \mid X, \widetilde{Y} - X] = \mathbb{E}[\varepsilon_Y \mid \widetilde{Y} - X] = \mathbb{E}[\varepsilon_Y \mid \varepsilon_1 + \varepsilon_Y], \tag{4.5}$$

where the second equality is due to the parametrization of $\widetilde{Y}$ and $X$; the third equality is due to $\varepsilon_Y \perp\!\!\!\perp X \mid \varepsilon_1 + \varepsilon_Y \implies \varepsilon_Y \perp\!\!\!\perp X \mid \widetilde{Y} - X$, and the last equality is due to Eq. (4.4).

Now, as our conditional expectation in Eq. (4.5) is shown to consist of terms entirely independent of $X$, we have that our predicted noise is independent of the un-noised conditioning variable:

$$\hat{\varepsilon}_Y \perp\!\!\!\perp X. \tag{4.6}$$

**Anticausal Direction: Denoising $\widetilde{X}$ and Testing Independence between $Y$ and $f_{\varepsilon_X}(\widetilde{X}, Y)$** In the anticausal direction, we repeat the same calculation and observe that the noise prediction is no longer independent of the input unnoised variable. First, substituting $Y = X + \varepsilon_1$ into $\widetilde{X} = X + \varepsilon_X$, we obtain

$$\widetilde{X} - Y = -\varepsilon_1 + \varepsilon_X. \tag{4.7}$$

We note that the same argument in the causal direction no longer works here as $\widetilde{X} - Y$ is not a sufficient statistic for $\mathbb{E}[\varepsilon_X \mid \tilde{X}, Y]$. In fact, we can show that

**Lemma 4.1.** $\hat{\varepsilon}_X = \mathbb{E}[\varepsilon_X | \widetilde{X}, Y] \not\perp\!\!\!\perp Y$.

The proof of Lemma 4.1 (Appendix E.11) proceeds by contradiction. While prior diffusion-based approaches have focused on leveraging diffusion to estimate the Jacobian of the score function [56], to our knowledge we are the first to point out an asymmetry arising from the independence of the predicted noise. Although the intuition is developed on a simple linear DGP, we hypothesize that the same argument generalizes to nonlinear DGPs, leading to more dependent predicted noise in the anticausal direction.

## 4.2 Theoretical Guarantees

Building on the intuition that we developed in Section 4.1, we build a decision rule that identifies the correct causal direction according to which denoising process (denoising $\widetilde{Y}$ or $\widetilde{X}$) leads to a prediction that is less dependent on the unnoised variable.

**Decision Rule 1** (Bivariate Denoising Diffusion (BiDD)). *Let* $\hat{\varepsilon}_Y = \varepsilon_{Y,\theta}(\widetilde{Y}, X)$, $\hat{\varepsilon}_X = \varepsilon_{X,\theta}(\widetilde{X}, Y)$ *be the predictions of the noise added to* $Y, X$, *respectively. Given a mutual information estimator* $\text{MI}(\cdot)$, *if* $\text{MI}(\hat{\varepsilon}_Y, X) < \text{MI}(\hat{\varepsilon}_X, Y)$, *conclude that* $X$ *causes* $Y$. *If* $\text{MI}(\hat{\varepsilon}_Y, X) > \text{MI}(\hat{\varepsilon}_X, Y)$, *conclude that* $X$ *causes* $Y$. *Else, do not decide.*

When the ANM-UM reduces to ANM (i.e., when Lemma 2.3 does hold), we can guarantee the correctness of Decision Rule 1 (Theorem 4.2, proof in Appendix E.12).

**Theorem 4.2** (Consistency of Decision Rule 1). *Suppose* $X, Y$ *follow Eq.* (2.1)*, Assumptions 2.1 and 2.2 hold, and no nonlinear mediator exists. Then, given a consistent mutual information estimator and infinite data, Decision Rule 1 correctly recovers the causal direction between* $X, Y$.

We note that, to our knowledge, Theorem 4.2 represents the first theoretical result on using denoising diffusion for bivariate causal discovery by leveraging asymmetries arising from the independence structure of ANMs. We discuss the major challenges in extending Theorem 4.2 to the irreducible case, i.e., when conditions described in Lemma 2.3 hold, in Appendix E.13.

Based on intuition from the ANM case, we conjecture that Decision Rule 1 remains consistent for cases when the ANM-UM does not reduce to ANM, such as when there is a nonlinear mediator. Note that as $X$ and $Y$ are functionally related, conditioning on either variable when predicting the added noise can induce dependence between the predicted noise and the conditioning variable in either direction. Under the ANM-UM, $Y$ can be modeled as a series of transformations of $X$ where at each step an independent error is injected additively. Intuitively, we hypothesize that the additive noise is easier to decouple from $Y$ when denoising $\tilde{Y}$ while conditioning on $X$, thereby rendering the estimate of the noise added to $Y$ ($\hat{\varepsilon}_Y$) less dependent on $X$. In contrast, in the anticausal direction, $X$ can only be modeled as a series of transformations of $Y$ where the error is not additively injected; this would imply that denoising $\tilde{X}$ while conditioning on $Y$ leads to a predicted noise that is more dependent on $Y$. Therefore, we focus on the gap between the dependencies in the two directions as the signal for inferring causal direction. We validate this conjecture empirically (Section 5), finding that our approach performs well across a wide variety of DGPs.

## 4.3 BiDD: A Practical Bivariate Denoising Diffusion Approach

Guided by the intuition developed in the linear case, we now present *BiDD*, a practical method for inferring causal direction based on asymmetries in the independence of denoising estimates.

BiDD fits two conditional diffusion models, one for each direction. For $B \to A$, we corrupt $A$ with noise and train to recover it given $B$, and vice versa. We then compare dependence between predicted noise and the condition, choosing the direction with lower dependence. We now describe each of these steps in detail. Additional information can be found in Appendix F, where we also formalize the procedure in Algorithm 1.

**Noise Prediction** We train a neural network to reconstruct the Gaussian noise injected into a noised sample $\tilde{A}_t$, conditioned on $B$. Our training follows the standard denoising diffusion framework of Ho et al. [15] and its conditional extensions [53].

Let $\{\alpha_t\}_{t \in [T]}$ denote a fixed noise schedule and let $\bar{\alpha}_t = \prod_{s=1}^{t} \alpha_s$ be its cumulative product. For a variable $A$ and a diffusion timestep $t$, we define the noised version:

$$\tilde{A}_t = \sqrt{\bar{\alpha}_t}\, A \; + \; \sqrt{1 - \bar{\alpha}_t}\, \varepsilon, \quad \varepsilon \sim \mathcal{N}(0, 1). \tag{4.8}$$

Given $(\tilde{A}_t, B, t)$, the model $\varepsilon_{A,\theta}$ is trained to minimize the noise prediction loss:

$$L_{\text{CDM}} = \mathbb{E}_{A,B,\varepsilon,t} \left[ \|\varepsilon - \varepsilon_\theta(\tilde{A}_t, B, t)\|^2 \right], \quad t \sim \text{Unif}(\{1, \dots, T\}).$$

At each iteration, we sample a timestep $t$, generate $\tilde{A}_t$, and update $\varepsilon_\theta$ by minimizing $L_{\text{CDM}}$ with stochastic gradient descent over $E$ epochs. This yields a trained model $\varepsilon_{A,\theta}$, which predicts $\hat{\varepsilon}_A = \varepsilon_{A,\theta}(\tilde{A}_t, B, t)$.

Note that our theoretical results in Section 4 assumes that, for each direction of denoising, the noised variable satisfies $\tilde{A} = A + \varepsilon$ with $A \perp\!\!\!\perp \varepsilon$, where $A$ denotes either $X$ or $Y$. In practice, *BiDD* uses the rescaled form $\tilde{A} = \sqrt{\bar{\alpha}_t}A + \sqrt{1 - \bar{\alpha}_t}\varepsilon$ to ensure that the forward process converges to a standard Gaussian, as is common in denoising diffusion models. However, our analysis still remains valid because independence is preserved under linear transformations.

**Dependence Testing**   After training, we evaluate the model on the test set. For each diffusion timestep $t = 1, \dots, T$, we generate noised inputs to estimate the dependence between the predicted noise and the conditioning variable. Specifically, for each test sample $(A_i, B_i)$, we construct $k$ noised versions $\{\tilde{A}_{t,1}^{(i)}, \dots, \tilde{A}_{t,k}^{(i)}\}$ by sampling independent noise $\varepsilon \sim \mathcal{N}(0, 1)$ $k$ times.

We then apply the trained model to obtain noise predictions $\hat{\varepsilon}_{A,i,j} = \varepsilon_{A,\theta}(\tilde{A}_{t,j}^{(i)}, B_i, t)$ for each noised sample. The mutual information $\text{MI}_{A,t}$ is computed between the predicted noises and the conditioning variable $B$ as $\text{MI}_{A,t} = \text{MI}(\hat{\varepsilon}_A, B)$.

We repeat the procedure in the reverse direction by training a second model that predicts $\hat{\varepsilon}_B$ from noised $B$ and conditioning on $A$, and compute $\text{MI}_{B,t}$ analogously.

**Inferring Causal Direction**   To determine the causal direction, we compare $\text{MI}_{A,t}$ and $\text{MI}_{B,t}$ for each timestep $t$. We count how often one direction yields a lower mutual information value and select the direction that does so more frequently across timesteps. A formal description of this procedure is provided in Subroutine 3 in Appendix G.2.

While our theoretical framework assumes sample splitting between training and testing, we find in practice that using the full dataset for both training and dependence estimation often improves performance, consistent with observations from prior work [18]. Therefore, we empirically evaluate two variants: $\text{BiDD}_{\text{Test}}$, which uses a held-out test set for dependence estimation, and $\text{BiDD}_{\text{Total}}$, which uses the full dataset. Additional implementation details, including learning rate, optimizer, noise schedule, and estimator configuration, are provided in Appendix G.2.

**Computational Complexity**   The computational complexity of BiDD involves two main stages. Firstly, the training of two conditional denoising diffusion models, each for $E$ epochs over $m_{\text{Train}}$ training samples. If $C_{\text{step}}$ denotes the cost of a single neural network training step (forward pass, loss computation, backward pass, and parameter update) per sample, this stage has a complexity of $O(E \cdot m_{\text{Train}} \cdot C_{\text{step}})$. Secondly, the inference stage as per Decision Rule 1 requires generating noise predictions for $l = k \cdot m_{\text{eval}}$ evaluation samples per timestep $T$ in the model (costing $O(m_{\text{eval}} \cdot k \cdot T \cdot C_{\text{fwd}})$, where $C_{\text{fwd}}$ is the neural network forward pass cost) and computing two mutual information (MI) estimates. If $C_{\text{MI},l}$ is the cost for one MI estimation on $l$ samples, this adds $C_{\text{MI},l}$ to the inference cost. The overall computational complexity of BiDD is thus $O(E \cdot m_{\text{Train}} \cdot C_{\text{step}} + T(m_{\text{eval}} \cdot k \cdot C_{\text{fwd}} + C_{\text{MI},l}))$, which is typically dominated by the $O(E \cdot m_{\text{Train}} \cdot C_{\text{step}})$ training component.

## 5   Experimental Results

We evaluate BiDD on synthetic data with linear, nonlinear invertible, and nonlinear non-invertible mechanisms, as well as a real-world dataset [55]. BiDD achieves state-of-the-art and consistent performance across settings, while most baselines performs poorly in at least one setting.

| Method Noise | Linear Unif. | Neural Net Gauss. | Neural Net Unif. | Quadratic Gauss. | Quadratic Unif. | Tanh Gauss. | Tanh Unif. |
|---|---|---|---|---|---|---|---|
| BiDD$_{\text{Total}}$ | 0.77 | .93 | .90 | 1.00 | 1.00 | 0.80 | .93 |
| BiDD$_{\text{Test}}$ | 0.73 | .97 | .97 | 1.00 | 1.00 | 0.60 | 0.83 |

Table 1: Accuracy of BiDD$_{\text{Total}}$ and BiDD$_{\text{Test}}$ across different transformation-noise combinations, with *no mediators* and $n = 1000$.

| Method Noise | Linear Unif. | Neural Net Gauss. | Neural Net Unif. | Quadratic Gauss. | Quadratic Unif. | Tanh Gauss. | Tanh Unif. |
|---|---|---|---|---|---|---|---|
| BiDD$_{\text{Total}}$ | 0.83 | 0.87 | 0.97 | **1.00** | **1.00** | 0.80 | 0.83 |
| BiDD$_{\text{Test}}$ | 0.80 | 0.87 | **1.00** | **1.00** | **1.00** | 0.63 | 0.77 |
| CANM | 0.10 | **0.93** | 0.87 | **1.00** | **1.00** | 0.50 | 0.10 |
| Adascore | 0.93 | 0.73 | 0.77 | 0.43 | 0.13 | 0.67 | **1.00** |
| NoGAM | **1.00** | 0.43 | 0.43 | 0.00 | **1.00** | 0.63 | **1.00** |
| SCORE | **1.00** | 0.73 | 0.57 | 0.43 | **1.00** | 0.50 | **1.00** |
| DagmaL | 0.13 | 0.17 | 0.10 | 0.00 | 0.00 | 0.00 | 0.07 |
| CAM | 0.03 | 0.77 | 0.80 | **1.00** | **1.00** | **0.93** | 0.13 |
| PNL | 0.73 | 0.83 | 0.70 | 0.83 | 0.83 | 0.67 | 0.70 |
| RESIT | 0.93 | 0.70 | 0.67 | **1.00** | **1.00** | 0.87 | **1.00** |
| DLiNGAM | **1.00** | 0.13 | 0.13 | 0.10 | 0.40 | 0.17 | **1.00** |
| Var-Sort | 0.43 | 0.57 | 0.63 | 0.47 | 0.57 | 0.33 | 0.60 |

Table 2: Accuracy of methods across different transformation-noise combinations, with *one mediator* and $n = 1000$. **Bold** indicates best, underline indicates second-best.

## 5.1 Setup

**Synthetic Data Details.** We produce synthetic bivariate causal pairs under the following ANM-UM (Eq 2.1), with varying causal mechanisms, exogenous noise distributions, sample size, and number of mediators. We use linear mechanisms with randomly drawn coefficients; we use both invertible (tanh) and non-invertible (quadratic, neural networks with randomly initialized weights [29, 23, 14]) nonlinear mechanisms. We use both uniform and Gaussian noise (excluding the linear Gaussian case to ensure identifiability). We standardize the data to mean 0 and variance 1 to ensure that the simulated data is sufficiently challenging; methods are evaluated on 20 randomly generated seeds in each experimental setting. See Appendix G.1.1 for details on parameters choice for each DGP.

**Real-World Data Details.** To confirm the real-world applicability of our approach, we test BiDD on the Tübingen Cause-Effect dataset [41], a widely used bivariate discovery benchmark that consists of 99 causal pairs that may have unobserved mediators. Due to runtime issues with baselines, we subsample the dataset of each causal pair by randomly selecting up to $n = 3000$ data points.

**Baselines and Evaluation.** We benchmark BiDD against a mix of classical and SOTA methods: we compare against three Residual-Independence methods (DirectLiNGAM, RESIT, PNL), three Score-Matching methods (SCORE, NoGAM, Adascore), two MSE-Minimization methods (DagmaLinear [3], CAM), and the only hidden mediator method in the literature (CANM). We include the heuristic algorithm Var-Sort, which exploits artifacts common to simulated ANMs [50], to show that BiDD performance is not driven by such shortcuts. Similar to [41], we use the *accuracy* for *forced decisions*, which corresponds to forcing the compared methods to decide the causal direction.

## 5.2 Results

**Synthetic Data.** We first examine the performance of BiDD$_{\text{Total}}$ in the traditional ANM setting (no unobserved mediator): Table 1 shows results for BiDD$_{\text{Total}}$ on data generated by different mechanism-noise combinations and sample size $n = 1000$. We observe the robust performance of BiDD$_{\text{Total}}$, achieving $\geq 77\%$ accuracy across all mechanisms. This empirically confirms the theoretical correctness guarantee given in Section 4 (Theorem 4.2).

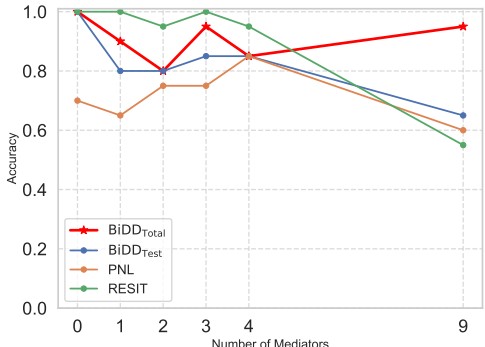
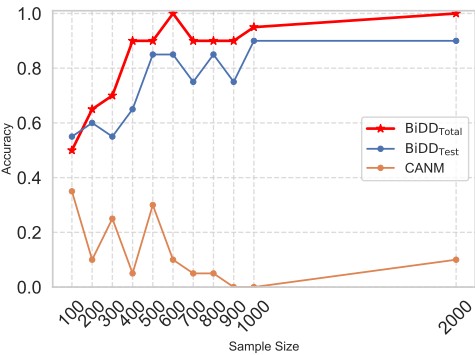

(a) $n = 1000$ with increasing number of mediators.

(b) One unobserved mediator, with increasing $n$.

Figure 2: Tanh mechanism, uniform noise setting.

| Method | $\text{BiDD}_{\text{Total}}$ | $\text{BiDD}_{\text{Test}}$ | CANM | CAM | Adascore | Entropy |
|---|---|---|---|---|---|---|
| **Accuracy** | 0.64 | 0.60 | 0.47 | 0.56 | 0.06 | 0.36 |
| **Method** | DagmaLinear | DirectLiNGAM | NoGAM | RESIT | PNL | SCORE |
| **Accuracy** | 0.30 | 0.51 | **0.69** | 0.62 | 0.61 | 0.65 |

Table 3: Accuracy for Tübingen Causal Pairs dataset, $n = 3000$

We now examine how $\text{BiDD}_{\text{Total}}$ performs under unmeasured mediators: in Table 2 we display results for different mechanism-noise combinations, each generated with one unobserved mediator (i.e., $Y = f_2(f_1(X) + \varepsilon_1) + \varepsilon_2$) and sample size $n = 1000$. We observe the robustness of $\text{BiDD}_{\text{Total}}$, as it achieves $\geq 80\%$ accuracy across all experimental setups, getting the first or second best accuracy $5/7$ times. In contrast, all baselines except PNL and RESIT perform extremely poorly ($\leq 50\%$) in at least two settings. PNL's performance is significantly lower ($\sim 10\% - 20\%$) than $\text{BiDD}_{\text{Total}}$ in almost every setting, while RESIT struggles in the neural network setting ($\leq 70\%$). The other hidden mediator method, CANM, performs poorly ($\leq 50\%$) for invertible mechanisms (linear, tanh), even for Gaussian noise, which is consistent with our analysis of CANM's behavior under posterior collapse (see Appendix E.9). The degraded baseline performance when the ANM assumption is violated highlights the limited applicability of current bivariate ANM methods. The performance gap becomes more pronounced when adding a second mediator (see Appendix H.3).

In Figure 2, we investigate how $\text{BiDD}_{\text{Total}}$ performs under fixed sample size ($n = 1000$) and varying depth (Figure 2a), or fixed depth (one mediator) and varying sample size (Figure 2a), in the tanh mechanism, uniform noise setting. In Figure 2a we observe that as the number of mediators increases, the performance of RESIT and PNL both degrade (to $\sim 60\%$), while $\text{BiDD}_{\text{Total}}$ remains performant ($\sim 95\%$). This shows that the performance of Residual-Independence based methods (RESIT and PNL) is sensitive to mediator number, while our diffusion approach remains robust. We observe a similar trend for the non-invertable case (see Appendix H.4)

In Figure 2b we observe the consistency of $\text{BiDD}_{\text{Total}}$, as its accuracy approaches $100\%$ while CANM does not improve, and in fact seems to decrease in performance. This points to CANM being inconsistent in settings with unmeasured mediators, rather than merely having finite sample issues.

**Real-world Data** The results of Tübingen dataset are presented in Table 3: $\text{BiDD}_{\text{Total}}$ ($64\%$) performs comparable to the best baselines, NoGAM ($69\%$) and SCORE ($65\%$), outperforming the rest of the methods. This confirms the robustness of BiDD across diverse real-world setups.

**Discussion.** Future work includes extending BiDD to be robust to latent confounding, and analyzing its potential consistency in settings where ANM-UM cannot be reformulated as an ANM.

## Acknowledgments

This work was supported in part by an AWS Cloud Credit Grant from Cornell's Center for Data Science for Enterprise and Society.

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

# Appendix

## A  Notation

| | |
|---|---|
| $\mathrm{Pa}(x_i)$ | The set of parent vertices of $x_i$. |
| $Z_i$ | The $i$'th Mediator |
| $f_i$ | Arbitrary function, generating $Z_i$ |
| $\varepsilon_i$ | An independent noise term sampled from an arbitrary distribution. |
| $X \perp\!\!\!\perp Y$ | $X$ is independent of $Y$ |
| $X \not\perp\!\!\!\perp Y$ | $X$ is not independent of $Y$ |
| $X \rightarrow Y$ | $X$ is a parent of $Y$ in the ANM-UM |
| $X \dashrightarrow Y$ | $X$ is a parent of $Y$ in the ANM-UM |
| $[T]$ | Set of integers $\{1, \ldots, T\}$ |
| $\{Z_i\}_{i \in [T]}$ | Set of Mediators $\{Z_1, \ldots, Z_T\}$ |
| $A_i$ | $i'$th data point |
| $\{A_i, B_i\}_{i \in [n]}$ | Collection of $n$ data points |
| $\mathbb{E}\,[Y|X]$ | Conditional expectation of $Y$ given $X$ |
| $f^{-1}$ | The inverse function of $f$ |
| $\nabla$ | Gradient operator |
| $Var[Y|X]$ | Variance of $Y$ given $X$ |
| $\tilde{X}$ | Noised version of $X$ |
| $f_{\varepsilon_X}$ | Predictor of the noise added to $X$ |
| $f_{\varepsilon_X}^{\star}$ | The best predictor of the noise added to $X$. Best means lowest MSE. |
| $MI(\cdot)$ | Empirical mutual-information estimator |
| $\varepsilon_A$ | Noise added to $\tilde{A} = A + \varepsilon_A$ |
| $\varepsilon_{A,\theta}$ | A learned model for predicting $\varepsilon_A$ with parameters $\theta$ |
| $O(\cdot)$ | Big-O (Landau) notation for asymptotic upper bound |
| $\exists$ | There exists |
| $\partial$ | Partial derivative |
| $\mathbb{E}_X[\cdot]$ | Expectation given $X$ |
| $T[\cdot]$ | Minimal-sufficient statistic |
| $\frac{\partial f}{\partial x}$ | Partial derivative with respect to $x$ |
| $\Delta f(u)$ | Difference of function shifted by a constant $c$ at $u$, $\Delta f(u) = f(u + x) - f(u)$ |
| $f_X$ | Probability density function of $X$ |
| $g_{X,Y}$ | Joint density of $X$ and $Y$ |
| $\sigma$ | Sigma algebra |
| $Cov(X, Y)$ | Covariance of $X$ and $Y$ |
| $||X||^2$ | Euclidian norm of $X$ |
| $\mathcal{N}(0, 1)$ | Normal distribution with mean 0 and variance 1 |
| $\mathcal{U}(-1, 1)$ | Continous uniform distribution between $-1$ and 1 |
| $\mathrm{Unif}(\{1, \ldots, N\})$ | Discrete uniform distribution between $N$ elements |
| $\mathbf{1}$ | Indicator function |
| $\mathcal{D}$ | Bivariate Dataset, consisting of $n$ observations of $A$ and $B$ |

# B  Further Discussion of Related Bivariate Methods

In this section, we further clarify the difference in modeling assumptions employed by ANM-based methods, LSNM-based methods, and ICM-based methods.

**Location Scale Noise Models.** LSNMs follow $Y = f(X) + g(X)\varepsilon$, where $X \perp\!\!\!\perp \varepsilon$, allowing the noise term $\varepsilon$ to be scaled and shifted for each $X$ value, according to $g(X)$. This increased flexibility models the possibility of heteroskedastic noise (noise dependent on the input $X$).

LSNMs are similar to ANM-UM, in that both include the vanilla ANM ($Y = f(X) + g(X)\varepsilon$) as a subcase. However, we note that while LSNMs and ANM-UM overlap, ANM-UM cover many cases which LSNMs do not. For example, $y = e^{x+\varepsilon_1} + \varepsilon_2$ can be modeled as a ANM-UM, while LSNMs do not admit such a representation. In general, ANM-UM can admit much more complicated joint distributions because they attempt to account for unobserved mediators: this introduces multiple (rather than one) independent noise distributions, with multiple transformations (rather than one) of the original input.

Additionally, methods developed to exploit LSNMs have several drawbacks. They either lack theoretical guarantees, or they require parametric assumptions than the general LSNM case. For example, [64] require linear mechanisms, while [18, 7] require Gaussian noise for correctness results.

**Principle of Independent Mechanism Approaches** The independence of cause and mechanism postulate [57] (ICM) states that the cause $X$ should be independent of the mechanism that maps $X$ to the effect $Y$. More concretely, this means that $X \dashrightarrow Y$ only if the shortest description of $P_{X,Y}$ is given by separate descriptions of $P_{Y|X}$ and $P_X$, in the sense that knowing $P_X$ does not enable a shorter description of $P_{Y|X}$ (and vice versa) [20]. Here description length is understand in the sense of algorithmic information ("Kolmogorov complexity") [25].

The overall ICM postulate is a true generalization of the ANM-UM (as well as the LSNM), as the functional mechanisms $f_1, \ldots, f_{T+1}$ and noise distributions $\varepsilon_1, \ldots, \varepsilon_{T+1}$ (which make up the data generating process from $X$ to $Y$) do not change for different input distributions of $X$. However, concrete methods developed to exploit ICM have many drawbacks. These generally follow from the fact that Kolmogorov complexity is known generally to not be computable [63]. Therefore, methods use approximations or proxies of the Kolmogorov complexity to develop heuristic approaches.

For example, [38] substitute the minimum message length principle, [62] use quantile scoring as a proxy for Kolmogorov complexity, and [32] leverage a condition on the parameter size of the true causal model implied by the Kolmogorov struction function. In general, methods based on the ICM do not come with strong identifiability results [32].

# C  Further Discussion of Related Global Methods

Recent work in global causal discovery has grappled with the issue of unobserved mediators, to various degrees of success. The multivariate version of Adascore [37] is the first score-matching method to handle unobserved confounders, but the authors clearly state that it fails to correctly recover the graph when unobserved mediators are present (See Examples 4, 5, and 7 in [37]). [31] showed that, under a further restriction of the ANM, where both the causal effects and error terms are additive (causal additive models, CAM, a subcase of ANM-UM), it is possible to recover the correct causal edge when all parents of a variable are measured, and otherwise leave the causal edge undecided if an unobserved mediator is a parent of an observed variable. In a recent extension of this work, Pham et al. [49] have shown that, under the CAM restriction, the correct causal edge can be recovered if the unmeasured mediator that is a parent of an observed variable is embedded in certain types of global graphical structures. However, neither of the latter two works comment on the general bivariate case involving unobserved mediators (AMM-UM), where additional global information may not be present.

Our bivariate method for handling unobserved mediators (BiDD) is motivated by the drawbacks of current ANM-based global discovery methods, as either they cannot handle unobserved mediators at all, fail to recover edges under hidden mediation, or can only do so under very narrow global graphical structures. Future work can incorporate BiDD as a subroutine in a global ANM-based discovery method, providing utility in real-world systems where hidden mediation abounds.

# D  Problem Setup

## D.1  Relation of ANM-UM to ANM, PNL, and CANM

Note that ANM-UM can be represented as $Y = f_{T+1}(f_T(\ldots) + \varepsilon_T) + \varepsilon_{T+1}$, where the last term inside the $\ldots$ is $f_1(X) + \varepsilon_1$.

The traditional ANM [47] models $Y = f(X) + \varepsilon_1$, where the key constraint is that $X \perp\!\!\!\perp \varepsilon_1$. If $T = 0$ for the ANM-UM, i.e., there are no unobserved mediators, then $f_1$ and $f_{T+1}$ coincide, and Eq 2.1 reduces to $Y = f_1(X) + \varepsilon_1$, which is exactly the ANM.

The PNL [67] models $Y = g(f(X) + \varepsilon_1)$ where $X \perp\!\!\!\perp \varepsilon_1$ and $g$ is an invertible nonlinear transformation. If $T = 1$ for the ANM-UM, $f_2$ is nonlinear and invertible, and $\varepsilon_{T+1} = 0$, then Eq 2.1 reduces to $Y = f_2(f_1(X) + \varepsilon_1)$, which is exactly the PNL.

The CANM [6] models $Y = f_{T+1}(f_T(\ldots) + \varepsilon_T) + \varepsilon_{T+1}$ where all $f_1, \ldots, f_{T+1}$ are nonlinear. If all $f_1, \ldots, f_{T+1}$ in Eq 2.1 are nonlinear, then ANM-UM reduces to the CANM.

## D.2  Identifiability Discussion

We first note that any ANM-UM (Eq 2.1) can be represented equivalently as

$$Y = F(X, \varepsilon_1, \ldots, \varepsilon_T) + \varepsilon_{T+1} \tag{D.1}$$

where $X, \varepsilon_1, \ldots, \varepsilon_{T+1}$ are all mutually independent, and $F = f_{T+1}(f_T(\ldots) + \varepsilon_T)$. For the ANM-UM to be identifiable, we require that there is no backwards ANM-UM that fits the anticausal direction $X \dashrightarrow Y$, i.e. there does not exist $G, \hat{\varepsilon}_1, \ldots, \hat{\varepsilon}_{T+1}$ such that

$$X = G(Y, \hat{\varepsilon}_1, \ldots, \hat{\varepsilon}_T) + \hat{\varepsilon}_{T+1} \tag{D.2}$$

where $G = g_{T+1}(g_T(\ldots) + \hat{\varepsilon}_T)$ and, additionally, $Y, \hat{\varepsilon}_1, \ldots, \hat{\varepsilon}_{T+1}$ are mutually independent. Theorem 1 from [6] shows that for any $G, \hat{\varepsilon}_1, \ldots, \hat{\varepsilon}_{T+1}$ which satisfy Eq D.2, we have that $Y, \hat{\varepsilon}_1, \ldots, \hat{\varepsilon}_{T+1}$ are mutually independent (and thus ANM-UM unidentifiable) if and only if, $\hat{\varepsilon}_{T+1}$ takes a very particular form:

$$p_{\hat{\varepsilon}_{T+1}}(\hat{\varepsilon}_{T+1}) = \int e^{2\pi i \hat{\varepsilon}_{T+1} \cdot \nu} \frac{\iint p(x)\, p(n)\, p_{\varepsilon_{T+1}}\big(y - f(x,n)\big)\, e^{-2\pi i\, x \cdot \nu}\, dn\, dx}{p(y) \int p(\hat{n})\, e^{-2\pi i\, g(y, \hat{n}) \cdot \nu}\, d\hat{n}}\, d\nu \tag{D.3}$$

where $n = \{\varepsilon_1, \ldots, \varepsilon_T\}$ and $\hat{n} = \{\hat{\varepsilon}_1, \ldots, \hat{\varepsilon}_T\}$

Therefore, to ensure identifiability, Assumption 2.2 requires that for any such $G, \hat{\varepsilon}_1, \ldots, \hat{\varepsilon}_{T+1}$ which satisfy Eq D.2, $\hat{\varepsilon}_{T+1}$ does not satisfy Eq D.3.

Cai et al. [6] show that, when ANM-UM can be represented as a traditional ANM (i.e., ANM-UM satisfies conditions in Lemma 2.3), the Assumption of 2.2 reduces to known identifiability constraints. For example, in Corollary 1 of Cai et al. [6], they show that if all mechanisms $f_1, \ldots, f_{T+1}$ in the ANM-UM are linear, then Assumption 2.2 reduces to requiring that at least one of $X, \{\varepsilon_i\}$ are non-Gaussian. This is exactly the constraint described by Shimizu et al. [58]. In Corollary 2 of Cai et al. [6], they show that if $T = 0$ in ANM-UM (no unobserved mediators), then Assumption 2.2 reduces to requiring that $X, f_1, \varepsilon_1$ satisfy the differential equation described in the identifiability assumptions of the general ANM in Hoyer et al. [16].

## D.3  Nonlinear ANM Not Always Closed Under Marginalization

Let $X \to Z_1 \to Y$, which corresponds to the ANM-UM $X, Z_1 = f_1(X) + \varepsilon_1, Y = f_2(Z_1) + \varepsilon_2$. As $Y = f_2(f_1(X) + \varepsilon_1) + \varepsilon_2$, it is straightforward that $Y$ follows an ANM if and only if $f_2(f_1(X) + \varepsilon_1)$ can be decomposed into the addition of a function of $X$ and a function of $\varepsilon_1$, i.e. $f_2(f_1(X) + \varepsilon_1) = A_1(X) + A_2(\varepsilon_1)$. Note that this follows the form of Pexider's equation $C(x + y) = D(x) + E(y)$ - it is known [1] that if $C, D, E$ satisfy this equation, $C, D, E$ must all be linear functions. Therefore, if $f_2$ is nonlinear, the ANM-UM does not reduce, and if $f_2$ is linear, then the ANM-UM does reduce.

### D.4 Proof of Lemma 2.3

**Lemma 2.3** (Irreducible ANM-UM and Nonlinear Mediator). *Under ANM-UM (Eq. (2.1)) and Assump.s 2.1 and 2.2, $Y$ does not admit a decomposition in Eq. (2.2) if and only if there exists a mediator $Z_i$ such that $Y = h(Z_i) + \tilde{\varepsilon}$ for some nonlinear $h$, with $\tilde{\varepsilon} \perp\!\!\!\perp Z_i$. Additionally, we call such a mediator $Z_i$ a* nonlinear mediator.

*Proof.* Suppose that there exists a mediator $Z_i$ such that $Y = h(Z_i) + \tilde{\varepsilon}$ where $h$ is nonlinear, and $Z_i \perp\!\!\!\perp \tilde{\varepsilon}$. Then, $\exists$ some function $F$ such that $Y = h(F(X, \varepsilon_1, \ldots, \varepsilon_{i-1}) + \varepsilon_i) + \tilde{\varepsilon}$. Suppose for contradiction that $Y$ admits an additive decomposition, i.e. $Y = h(F(X, \varepsilon_1, \ldots, \varepsilon_{i-1}) + \varepsilon_i) + \tilde{\varepsilon} = A_1(X) + A_2(\varepsilon_1, \ldots, \varepsilon_i) + A_3(\tilde{\varepsilon})$. Note that this implies $h(F(X, \varepsilon_1, \ldots, \varepsilon_{i-1}) + \varepsilon_i) + \tilde{\varepsilon} = A_1(X) + A_2(\varepsilon_1, \ldots, \varepsilon_i)$, which follows the form of Pexider's equation, $C(x + y) = D(x) + E(y)$. It is known that if $C, D, E$ satisfy this equation, $C, D, E$ must all be linear functions [1]. However, this contradicts that $h$ is nonlinear. Therefore, $Y$ does not admit a decomposition in Eq 2.2.

Suppose that $Y$ does not admit a decomposition in Eq 2.2. Suppose for contradiction that there does not exist a nonlinear mediator $Z_i$. This implies that $f_2, \ldots, f_{T+1}$ are all linear functions. Then, $Y$ can be written as a linear function of $Z_1$ and noise terms $\varepsilon_2, \ldots, \varepsilon_{T+1}$, i.e. $Y = \alpha Z_1 + \sum_{i=2}^{T+1} \beta_i \varepsilon_i$. Then, $Y = \alpha f_1(X) + \alpha \varepsilon_1 + \sum_{i=2}^{T+1} \beta_i \varepsilon_i$ which follows the additive decomposition in Eq 2.2. Therefore, there must exist a nonlinear mediator $Z_i$.

$\square$

## E  Failure-mode of Prior Work + Proof of Lemma 4.1, Theorem 4.2, Experimental Analysis of CANM

### E.1  Decision Rules

**Decision Rule E.2** (Regression Residual-Independence). *Let $e_1, e_2$ be the residuals obtained from regressing $Y$ onto $X$, and $X$ onto $Y$ (respectively). If $e_1 \perp\!\!\!\perp X, e_2 \not\perp\!\!\!\perp Y$, then conclude $X$ causes $Y$. If $e_1 \not\perp\!\!\!\perp X, e_2 \perp\!\!\!\perp Y$, then conclude $Y$ causes $X$. If $e_1 \perp\!\!\!\perp X, e_2 \perp\!\!\!\perp Y$, conclude neither causes each other. Otherwise, do not decide.*

**Decision Rule E.3** (Nonlinear ICA Residual-Independence). *Check to see if the hypothesis $X \dashrightarrow Y$ holds and the hypothesis $Y \dashrightarrow X$ holds. If only one hypothesis holds, we conclude that one is the causal direction. If they both hold, conclude there is no causal relationship. Otherwise, do not decide.*

**Decision Rule E.4** (Adascore Score-matching). *Let $r_1, r_2$ be the residuals obtained from regressing $Y$ onto $X$, and $X$ onto $Y$ (respectively). Let $s_1, s_2$ be the values obtained by plugging $r_1, r_2$ into $\mathbb{E}\left[ \left( \mathbb{E}\left[ \partial_Y \log p(X, Y) \mid r_1 \right] - \partial_Y \log p(X, Y) \right)^2 \right]$ and $\mathbb{E}\left[ \left( \mathbb{E}\left[ \partial_X \log p(X, Y) \mid r_2 \right] - \partial_X \log p(X, Y) \right)^2 \right]$ respectively. If $s_1 = 0, s_2 \neq 0$, conclude that $X \dashrightarrow Y$. If $s_1 \neq 0, s_2 = 0$, conclude that $Y \dashrightarrow X$. Else, do not decide.*

**Decision Rule E.5** (MSE-Minimization). *Let $Loss_{X \dashrightarrow Y}, Loss_{Y \dashrightarrow X}$ be the MSE obtained from predicting $Y$ from $X$, and $X$ from $Y$ respectively. Then if $Loss_{X \dashrightarrow Y} < Loss_{Y \dashrightarrow X}$, conclude that $X \dashrightarrow Y$. If $Loss_{X \dashrightarrow Y} > Loss_{Y \dashrightarrow X}$, conclude that $X \dashrightarrow Y$. Otherwise, do not decide.*

**Decision Rule E.6** (ELBO-Maximization). *Let $ELBO_{X \dashrightarrow Y}, ELBO_{Y \dashrightarrow X}$ be the ELBOs obtained from training a VAE to predict $Y$ from $X$, and $X$ from $Y$ respectively. Then if $ELBO_{X \dashrightarrow Y} > ELBO_{Y \dashrightarrow X}$, conclude that $X \dashrightarrow Y$. If $ELBO_{X \dashrightarrow Y} < ELBO_{Y \dashrightarrow X}$ conclude that $Y \dashrightarrow X$. Otherwise, do not decide.*

### E.2  Decision Rule Discussion

We note that while the Decision Rules E.2-E.6 are generally representative of each type of ANM-based bivariate methods (Regression Residual-Independence, Score-Matching, MSE-Minimization, etc.), there exist subclasses of methods in each category. For example, while Decision Rule E.4 reflects the method Adascore [37] (and NoGAM [35] to some extent), the score-matching method SCORE [52] leverages a slightly different condition on the score function to recover the causal direction. In our analysis, we choose each Decision Rule to reflect the methodology of the most

general (and typically most recent) method developed in each category. For example, we choose to focus on Adascore over SCORE, as Adascore handles linear, nonlinear, and non-Gaussian ANM, while SCORE requires nonlinear Gaussian ANM.

### E.3 Proof that Regression Residual-Independence Fails (Lemma 3.1)

**Lemma 3.1** (Regression Residual-Independence Fails). *Assuming a consistent estimator for regression residuals and access to infinite data, Decision Rule E.2 fails to identify the correct causal direction when at least one mediator is nonlinear.*

*Proof.* We note that Decision Rule E.2 identifies the causal direction if and only if the residual obtained from regressing $Y$ onto $X$ is independent of $X$, i.e. $e_1 \perp\!\!\!\perp X$. Suppose for contradiction that $e_1 \perp\!\!\!\perp X$. Then, we have that for $e_1 = Y - g(X) = h(\varepsilon_1, \ldots, \varepsilon_{T+1})$, where $g(X) = \mathbb{E}[Y|X]$, $e_1 \perp\!\!\!\perp X$. Therefore, we can rewrite $Y = g(X) + h(\varepsilon_1, \ldots, \varepsilon_{T+1})$. However, this leads to a contradiction: if there is at least one nonlinear mediator, then by Lemma 2.3, the ANM-UM underlying $X, Y$ does not admit an additive decomposition. Instead, $Y = A_1(X) + A_2(\varepsilon_{T+1}) + A_3(X, \varepsilon_1, \ldots, \varepsilon_T)$, where $A_3(X, \varepsilon_1, \ldots, \varepsilon_T)$ contains nonlinear interaction between $X$ and noise terms $\varepsilon$. Therefore, $e_1 \not\perp\!\!\!\perp X$, and therefore Decision Rule E.2 fails to identify the causal direction. $\square$

### E.4 Discussion of Residual-Dependence Comparisons

We note that despite Decision Rules E.2 being framed in terms of the outcomes of independence tests, most implementations of Residual-Dependence tests leverage the comparison of test statistic values that measure dependence, rather than strictly comparing the outcome of independence tests. For example, DirectLiNGAM [58] estimates and compares the mutual information, while RESIT [47] estimates and compares the $p$-value of the HSIC independence test ([10]. We analyze the empirical performance of such an approach in Section 5, as the baselines we use (DirectLiNGAM, RESIT) leverage these dependence comparisons to boost performance.

### E.5 Proof that Post-Nonlinear Residual Independence Fails (Lemma 3.2)

**Lemma 3.2** (PNL Residual-Independence Fails). *Assuming a consistent ICA residual estimator and access to infinite data, Decision Rule E.3 fails to recover the correct causal direction when there exists at least one non-invertible nonlinear mediator.*

*Proof.* As there exists at least one non-invertible nonlinear mediator (i.e., $\exists$ a function $f_t$ where $t \geq 2$ and $f_t$ non-invertible and nonlinear in the ANM-UM generating $Y$ from $X$), we note that by Lemma 2.3 we can rewrite $Y$ as $Y = A_1(X) + A_2(\varepsilon_1, \ldots, \varepsilon_{T+1}) + A(f_1(X) + \varepsilon_1, \ldots, \varepsilon_{T+1})$, where $A_3(\cdot)$ produces nonlinear interaction between $X$ and $\varepsilon$, and $A_3(\cdot)$ is non-trivial (non-zero), and non-invertible in $f_1(X) + \varepsilon_1$.

We note that Decision Rule E.3 identifies the causal direction if and only if the causal hypothesis $X \dashrightarrow Y$ holds, and the hypothesis $Y \dashrightarrow X$ does not hold. We note that the causal hypothesis $X \dashrightarrow$ holds if and only if $\exists$ functions $l_1, l_2$ such that for $e_1 = l_2(Y) - l_1(X)$, $e_1 \perp\!\!\!\perp x$. Suppose for contradiction that $\exists$ functions $l_1, l_2$ such that for $e_1 = l_2(Y) - l_1(X)$, $e_1 \perp\!\!\!\perp x$. Note that $e_1$ must be some function of noise terms $\varepsilon$, i.e., $e_1 = h(\varepsilon_1, \ldots, \varepsilon_{T+1}) = h(\varepsilon)$. Note that $l_2$ must be invertible, as otherwise it would contradict that $Y$ is a proper function of $X$ and noise terms $\varepsilon$, as there exists an original DGP $Y = F(X, \varepsilon_1, \ldots, \varepsilon_{T+1})$.

Suppose $l_2^{-1}$ is linear. Then, we can write $Y = \alpha(l_1(X)) + \alpha(e_1)$. However, this contradicts the non-triviality of $A_3(\cdot)$. Then $l_2^{-1}$ must be nonlinear and invertible. However, that contradicts the fact that $A_3(\cdot)$ is non-invertible in $f(X) + \varepsilon_1$. Therefore, there cannot exist functions $l_1, l_2$ such that for $e_1 = l_2(Y) - l_1(X)$, $e_1 \perp\!\!\!\perp x$. Therefore Decision Rule E.2 fails to identify the causal direction. $\square$

### E.6 Proof Score-Matching Fails (Lemma 3.3)

**Lemma 3.3** (Score-Matching Fails). *Assuming a consistent estimator of the conditional expectation and access to infinite data, Decision Rule E.4 fails to recover the correct causal direction when there exists at least one nonlinear mediator.*

*Proof.* We note that for Decision Rule E.4 to correctly identify the causal direction, it requires that $E\left[\left(E\left[\partial_Y \log p(X,Y) \,|\, r_1\right] - \partial_Y \log p(X,Y)\right)^2\right] = 0$. Notably, [37] shows (Proposition 4) that this holds if and only if for $Y = A_1(X) + A_2(U)$, we have $A_2(U) \perp\!\!\!\perp X$. However, as there is at least one nonlinear mediator, then by Lemma 2.3, the ANM-UM underlying $X, Y$ admits the following decomposition $Y = A_1(X) + A_2(\varepsilon_{T+1}) + A_3(X, \varepsilon_1, \ldots, \varepsilon_T)$, where $A_3(X, \varepsilon_1, \ldots, \varepsilon_T)$ contains nonlinear interaction between $X$ and noise terms $\varepsilon$, and is non-trivial. Therefore, by it follows from Proposition 4 of [37] that $E\left[\left(E\left[\partial_Y \log p(X,Y) \,|\, r_1\right] - \partial_Y \log p(X,Y)\right)^2\right] \neq 0$, and therefore Decision Rule E.4 fails to recover the right causal direction. In fact, [37] explicitly states that their method fails to recover causal relationships when unobserved mediators occur (see Examples 4, 5, 7 in [37]). $\square$

### E.7 Proof that MSE-Minimization Fails (Lemma 3.4)

**Lemma 3.4** (MSE-Minimization Fails). *Given a consistent estimator of the conditional expectation and infinite data, Rule E.5 fails to recover the correct causal direction when $\mathbb{E}[Var[X|Y]] < \mathbb{E}[Var[Y|X]]$.*

*Proof.* We note that under infinite data the optimal estimator of the MSE

$$\text{MSE}(f) = \mathbb{E}[(Y - f(X))^2],$$

converges to the conditional expectation

$$f^*(X) = \mathbb{E}[Y \mid X].$$

This implies that as the sample size $n$ goes to infinity, the MSE converges to the expected conditional variance of $Y|X$:

$$\begin{aligned}
\mathbb{E}_X[\text{MSE}(f)] &= \mathbb{E}_X[(Y - f^*(X))^2 \mid X] \\
&= \mathbb{E}_X[(Y - \mathbb{E}[Y \mid X])^2 \mid X] \\
&= \mathbb{E}_X[\text{Var}(Y \mid X)]
\end{aligned}$$

Therefore, if $\mathbb{E}[Var[X|Y]] < \mathbb{E}[Var[Y|X]]$, this implies that $Loss_{X \dashrightarrow Y} > Loss_{Y \dashrightarrow X}$, which implies that Decision Rule E.5 fails to recover the causal direction.

We note that $\mathbb{E}[Var[X|Y]] < \mathbb{E}[Var[Y|X]]$ can occur if the $R^2$-sortability favors the anti-causal direction. We note that the coefficient of determination $R^2$ is a simple function of the expected conditional variance, when the variables are standardized:

$$\begin{aligned}
R^2_{X \to Y}(f^*) &= 1 - \frac{\mathbb{E}[(Y - f^*)^2]}{\text{Var}(Y)} = 1 - \mathbb{E}_X[\text{Var}(Y \mid X)] \\
R^2_{Y \to X}(f^*) &= 1 - \frac{\mathbb{E}[(X - f^*)^2]}{\text{Var}(X)} = 1 - \mathbb{E}_Y[\text{Var}(X \mid Y)]
\end{aligned}.$$

Reisach et al. [51] show that linear ANM may or may not always be $R^2$-sortability; as linear ANM are a subset of ANM-UM, this justifies our claim that MSE-Minimization methods will fail on some ANM-UM. $\square$

### E.8 Necessary Constraints for MSE-Minimization methods

We note that the conditions under which MSE-Minimization actually does actually correspond to causal direction identification, i.e., the conditions that ensure $\mathbb{E}[Var[X|Y]] > \mathbb{E}[Var[Y|X]]$, have been discussed in other work. For example, [4] show that under the assumption of independence between the function relating cause and effect, the conditional noise distribution, and the distribution of the cause, as well as a close to deterministic causal relation, the errors are smaller in the causal direction. Marx & Vreeken [32] build up on Bloebaum et al. [4], and show that, under the assumption that the best anti-causal model requires at least as many parameters as the causal model (leveraging Kolmogorv's structure function), the regression errors should be smaller in the causal direction. However, these assumptions are quite distinct from our ANM-UM setting, and we leave further investigation to future work.

### E.9 Failure Mode of VAE

CANM uses a variational autoencoder (VAE) framework to decide causal direction by picking the direction with the lower ELBO. The training objective used consists of three parts: the log likelihood of $x$ (which does not depend on the model parameters $\theta$, the KL divergence of the latent code, and the reconstruction error [6, Equation 4].

Which of these terms dominates the loss is highly dependent on the training procedure for the VAE, since training VAEs is known to suffer from posterior collapse [11], which we observed during running the method. In Figure 3, we plot a decomposition of the training loss of the VAE for CANM, consisting of the KL-divergence term in the latent space and the reconstruction error, for different training regimes. Under the standard training setup (Figure 3a), the KL divergence remains close to zero, while the reconstruction error is relatively high, indicating that the model relies only on $X$ to reconstruct $Y$, effectively ignoring the latent representation.

Different mitigation strategies for mitigating posterior collapse have been proposed in the literature. Among them, the line following $\beta$-VAE, which introduces a factor in front of the $KL$ term, and uses a scheduling of the $\beta$ part during training [9, 12].

In our experiments, we found that training the VAE with a cyclical $\beta$ annealing schedule, following Fu et al. [9], led to better reconstruction results. The corresponding loss decomposition is shown in Figure 3b. Compared to the constant $\beta$ setting, the reconstruction error is lower, indicating that the model reconstructs the noise effectively. This improvement comes at the cost of a higher KL divergence penalty, suggesting that the latent space is being utilized more meaningfully.

Comparing the two schedules, Figure 3 shows that the dominant loss term varies with the training procedure: with a constant $\beta$, the reconstruction error dominates, while with a cyclical $\beta$, the KL divergence becomes more prominent.

As shown in Table 4, cyclical scheduling failed to improve performance for the invertible cases (*tanh* and *linear*) under uniform noise. The algorithm predicts the *opposite* causal direction with high probability ($\geq 90\%$). While achieving a better reconstruction after improved training, the accuracy in the *tanh + Gaussian* setting declined using the new training schedule. Both phenomena imply that it is exploiting a heuristic signal rather than truly recovering the correct causal orientation.

| Method | Linear | Neural Net | | Quadratic | | Tanh | |
| Noise | Unif. | Gauss. | Unif. | Gauss. | Unif. | Gauss. | Unif. |
|---|---|---|---|---|---|---|---|
| CANM | **0.10** | 0.93 | **0.87** | **1.00** | **1.00** | 0.50 | **0.10** |
| CANM (constant $\beta$) | 0.00 | **0.97** | 0.83 | 1.00 | 0.90 | **0.83** | **0.10** |

Table 4: Accuracy of CANM variants across different transformation–noise combinations, with *one mediator* and $n = 1000$. **Bold** indicates best.

The original CANM paper proposes selecting the number of latent variables by comparing model likelihoods across different dimensionalities. In our implementation, we instead provide CANM with the ground-truth number of mediators as a hyperparameter, which defines the size of its latent space. In contrast, BiDD does *not* require knowledge of the number of mediators.

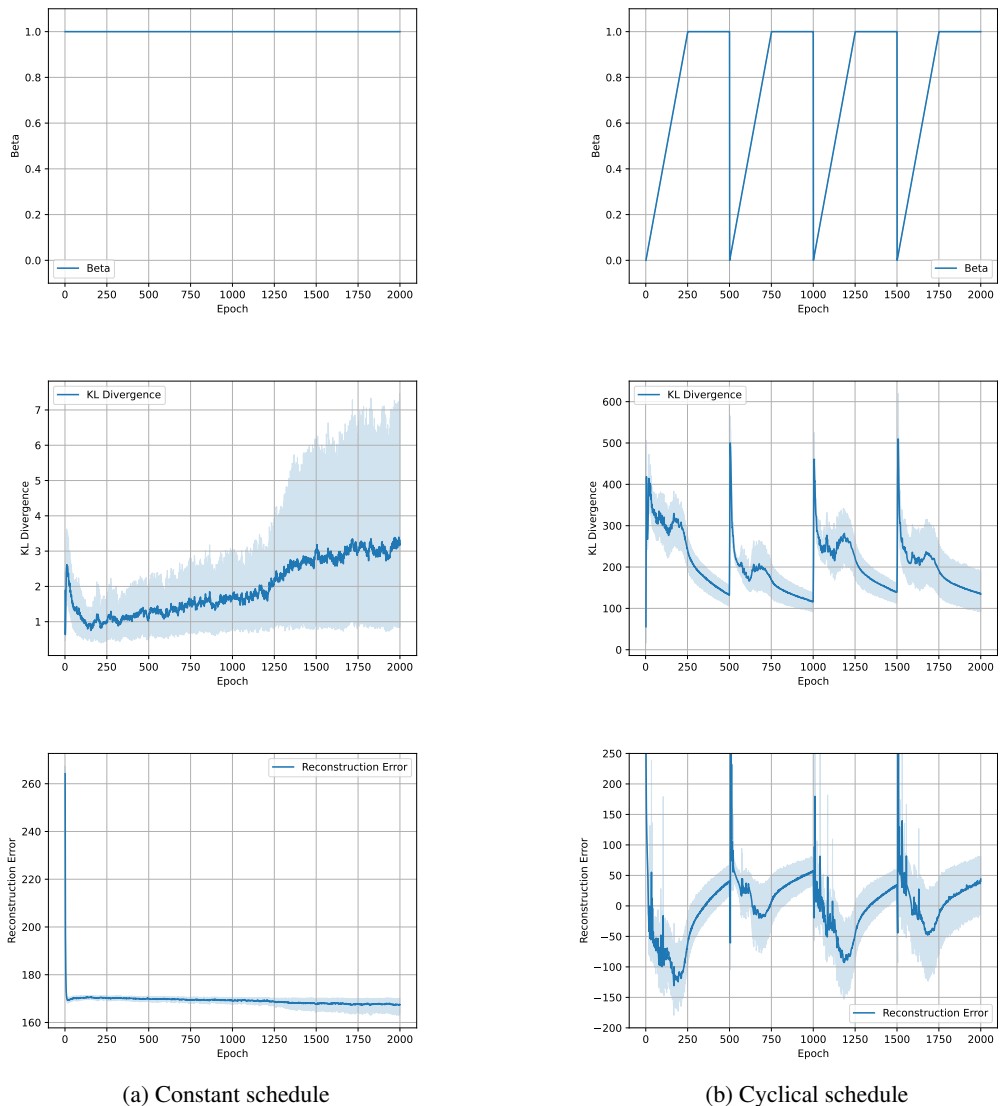

(a) Constant schedule          (b) Cyclical schedule

Figure 3: Comparison between different training schedules for $\beta$ in adapted $\beta$-VAE for the tanh uniform noise case. From top to bottom: $\beta$(t), KL divergence, example reconstructions. Mean and 95% bootstrap confidence interval (20 runs) from estimating $\varepsilon_Y$ from $\tilde{Y}$ and $X$.

### E.10    Proof that CANM Fails (Lemma 3.5)

**Lemma 3.5** (CANM Fails). *Assuming infinite data and a consistent estimator of the conditional expectation, Rule E.6 fails to recover the causal direction if posterior collapse occurs and the expected conditional log-likelihood minus the entropy is higher in the causal direction.*

*Proof.* We note that Decision Rule E.6 identifies the causal direction when posterior collapse occurs if and only if that expected conditional log-likelihood minus the entropy is lower in the causal direction. As we assume it is higher in the causal direction, this implies that Decision Rule E.6 must fail. $\square$

### E.11    Proof of Lemma 4.1 - Anticausal Direction in Linear ANM

**Lemma 4.1.** $\hat{\varepsilon}_X = \mathbb{E}[\varepsilon_X | \widetilde{X}, Y] \not\perp\!\!\!\perp Y.$

*Proof.* This proof proceeds through the following steps:

1. First, we will restate the DGP of $X, Y, \varepsilon_X, \tilde{X}$, and all assumptions.

2. We define the regression function $h(\tilde{X}, Y) := \mathbb{E}[\varepsilon_X | \tilde{X}, Y]$. This proof will proceed in the following steps.

   (a) We will characterize $h(\tilde{X}, Y)$, splitting it up into univariate functions and functions with interaction terms.
   (b) We will outline all possible cases (and subcases) for how the functional form of $h(\tilde{X}, Y)$ might look.
   (c) For each case (and associated subcases) we will show that $h(\tilde{X}, Y)$ is a function of a noise term dependent on $Y$.
   (d) We conclude that, as $h(\tilde{X}, Y)$ is always a non-trivial function of noise that is dependent on $Y$, we have that $h(\tilde{X}, Y) \not\perp\!\!\!\perp Y$.

**Step 1: Restate DGP and Assumptions**

Note, $X, Y$ follow

$$Y = X + \varepsilon_1, \ X \perp\!\!\!\perp \varepsilon_1. \tag{E.1}$$

We require Assumption 2.1 and 2.2); note that 2.2) stipulates that $\varepsilon_1$ is non-Gaussian, for identifiability. We inject independent Gaussian noise into both $X$, obtaining the noised term $\tilde{X}$:

$$\widetilde{X} = X + \varepsilon_X \ \ \varepsilon_X \sim \mathcal{N}(0, 1). \tag{E.2}$$

Additionally, we require that $X, \varepsilon_1, \varepsilon_X$ are random variables with everywhere-positive, absolutely continuous and differentiable densities $f_X$, $f_{\varepsilon_1}$, and $f_{\varepsilon_X}$.

**Step 2: Structure of $h(\tilde{X}, Y)$**

Note that $h(\tilde{X}, Y)$ can be decomposed as $h(\tilde{X}, Y) = A_1(\tilde{X}) + A_2(Y) + A_3(\tilde{X}, Y)$, where $A_3$ contains only (linear or nonlinear) interaction between $\tilde{X}$ and $Y$, while $A_1, A_2$ are univariate.

**Step 3: General Cases**

We note that the functional form of $h(\tilde{X}, Y)$ can be divided into four general cases: 1) $A_3$ is non-trivial, 2) $A_3$ and $A_2$ are trivial, $A_1$ is non-trivial, 3) $A_3$ and $A_1$ are trivial, $A_2$ is non-trivial, and 4) $A_3$ is trivial, while $A_1$ and $A_2$ are non-trivial.

Case 1): Suppose $A_3$ is non-trivial. This implies that there exist interaction terms between $\tilde{X}, Y$. Suppose for contradiction that $h(\tilde{X}, Y)$ is not dependent on a noise term dependent on $Y$. Then, it implies that $A_1, A_2$ somehow cancel out all $Y$ terms in $A_3$. This leads to a contradiction, since the space of additive functions—i.e., those expressible as a linear combination of univariate functions of each variable—cannot represent interaction terms, which require non-additive combinations such as products of variables. Therefore, $h(\tilde{X}, Y)$ is a non-trivial function of $Y$, making it a non-trivial function of noise term $\varepsilon_1, \varepsilon_1 \not\perp\!\!\!\perp Y$.

Case 2): Suppose $A_3, A_2$ are trivial, and $A_1$ is non-trivial. Then, for some function $g$, $h(\tilde{X}, Y) = g(\tilde{X}) = g(X + \varepsilon_X)$. This implies that $h(\tilde{X}, Y)$ is a non-trivial function of $X$ and a noise jointly independent of $X$ and $Y$. As $Y$ contains $X$ as a noise term ($Y = X + \varepsilon_1$), this implies that $Y \not\perp\!\!\!\perp h(\tilde{X}, Y)$.

Case 3): Suppose $A_3, A_1$ are trivial, and $A_2$ is non-trivial. This directly implies that $h(\tilde{X}, Y)$ is a non-trivial function of $Y$. Therefore, $Y \not\perp\!\!\!\perp h(\tilde{X}, Y)$.

Case 4): Suppose $A_3$ is trivial, while $A_1$ and $A_2$ are non-trivial. We analyze what happens here in Step 4 , further splitting Case 4 into subcases based on the linearity/nonlinearity of $A_1, A_2$.

**Step 4: Analyzing Case 4**

We split Case 4 into the following three subcases: A) $A_1$ is nonlinear, B) $A_2$ is nonlinear, C) both $A_1, A_2$ are linear.

Subcase A): Suppose $A_1(\tilde{X})$ is nonlinear. Then, as $\tilde{X} = X + \varepsilon_X$, $A_1(\tilde{X})$ will contain an interaction term between $X, \varepsilon_X$. Note as $A_3$ is trivial, the interaction term between $X, \varepsilon_X$ cannot be cancelled out - therefore, $h(\tilde{X}, Y)$ must be a non-trivial function of $X$. As $Y = X + \varepsilon_1$, this implies that $Y \not\perp\!\!\!\perp h(\tilde{X}, Y)$.

Subcase B): Suppose $A_2(Y)$ is nonlinear. Then, as $Y = X + \varepsilon_1$, $A_2(Y)$ will contain an interaction term between $X, \varepsilon_1$. Note as $A_3$ is trivial, the interaction term between $Y, \varepsilon_1$ cannot be cancelled out - therefore, $h(\tilde{X}, Y)$ must be a non-trivial function of $\varepsilon$ and $Y$. As $Y = X + \varepsilon_1$, this implies that $Y \not\perp\!\!\!\perp h(\tilde{X}, Y)$.

Subcase C): Suppose $A_1$ and $A_2$ are both linear, i.e.

$$A_1(\tilde{X}) = \alpha X + \alpha \varepsilon_X \tag{E.3}$$
$$A_2(Y) = \beta Y. \tag{E.4}$$

Suppose for contradiction that $h(\tilde{X}, Y) \perp\!\!\!\perp Y$. Then, we have

$$h(\tilde{X}, Y) = A_1(\tilde{X}) + A_2(Y) \tag{E.5}$$
$$h(\tilde{X}, Y) = \alpha X + \alpha \varepsilon_X + \beta Y. \tag{E.6}$$
$$\implies X = \frac{h(\tilde{X}, Y) - \beta Y - \alpha \varepsilon_X}{\alpha} \tag{E.7}$$
$$X = -\frac{\beta}{\alpha} Y + \frac{h(\tilde{X}, Y) - \alpha \varepsilon_X}{\alpha}. \tag{E.8}$$

Note that $h(\tilde{X}, Y) \perp\!\!\!\perp Y, Y \perp\!\!\!\perp \varepsilon_X$. Additionally, when $-\varepsilon_X$ is added to $\frac{h(\tilde{X}, Y)}{\alpha}$, it cannot cancel out any term dependent on $Y$. Therefore, the sum must be independent of $Y$, i.e. $Y \perp\!\!\!\perp \frac{h(\tilde{X}, Y) - \alpha \varepsilon_X}{\alpha}$. However, this contradicts our assumption in Step 1 that there does not exist a backwards ANM model $X = Y + n$ where $Y \perp\!\!\!\perp n$. Thus, $h(\tilde{X}, Y) \not\perp\!\!\!\perp Y$.

**Step 5: Conclusion**

We have shown that $h(\tilde{X}, Y)$ is a non-trivial function of $Y$ for all possible forms that $h(\tilde{X}, Y)$ can take. Therefore, we conclude that $\mathbb{E}[\varepsilon_X | \tilde{X}, Y] \not\perp\!\!\!\perp Y$.

$\square$

### E.12  Proof of Diffusion Correctness for ANM

**Theorem 4.2** (Consistency of Decision Rule 1). *Suppose $X, Y$ follow Eq. (2.1), Assumptions 2.1 and 2.2 hold, and no nonlinear mediator exists. Then, given a consistent mutual information estimator and infinite data, Decision Rule 1 correctly recovers the causal direction between $X, Y$.*

*Proof.* We note that if no nonlinear mediator exists, then by Lemma 2.3 $X, Y$ can be represented by a standard ANM, where $Y = f(X) + \varepsilon_1, X \perp\!\!\!\perp \varepsilon_1$ and Assumptions 2.1, 2.2 still hold. We additionally require that $X, \varepsilon_1$ are random variables with everywhere-positive, absolutely continuous densities $f_X, f_{\varepsilon_1}$, and $f_{\varepsilon_X}$. We further assume that $f$ is continuous and three-times differentiable. Note again, that we inject independent Gaussian noise into both $X$ and $Y$, obtaining the noised terms

$$\widetilde{X} = X + \varepsilon_X \quad \text{and} \quad \widetilde{Y} = Y + \varepsilon_Y, \quad \varepsilon_X, \varepsilon_Y, \sim \mathcal{N}(0, 1). \tag{E.9}$$

We note that, under infinite data, Decision Rule 1 correctly recovers the causal direction if and only if both of the following statements hold(written equivalently in terms of mutual information and independence):

$$MI(\hat{\varepsilon}_Y, X) = 0 \iff \mathbb{E}[\varepsilon_Y |, \tilde{Y}, X] \perp\!\!\!\perp X \tag{E.10}$$
$$MI(\hat{\varepsilon}_X, Y) > 0 \iff \mathbb{E}[\varepsilon_X |, \tilde{X}, Y] \not\perp\!\!\!\perp Y \tag{E.11}$$

We will first show the causal direction (Eq E.10 holds), then the anticausal direction (Eq E.11 holds).

### E.12.1 Causal Direction

We will find a statistic $T$ such that $\mathbb{E}[\varepsilon_Y | \tilde{Y}, X] = \mathbb{E}[\varepsilon_Y | T]$, and $X$ is jointly independent of $T$ and $\varepsilon_Y$. Then, we will conclude that Eq E.10 holds.

Note, the joint density $g_{\tilde{Y}, X, \varepsilon_Y}$ can be written as

$$g_{\tilde{Y}, X, \varepsilon_Y}(\tilde{y}, x, e) = f_X(x) f_{\varepsilon_1}(\tilde{y} - f(x) - e) f_{\varepsilon_Y}(e) \tag{E.12}$$

using the change of variables

$$\tilde{Y} = f(X) + \varepsilon_1 + \varepsilon_Y, \tag{E.13}$$

as $X, \varepsilon_1, \varepsilon_Y$ are mutually independent.

Let $T(\tilde{Y}, X) = \tilde{Y} - f(X)$. Then, we have that

$$g_{\tilde{Y}, X, \varepsilon_Y}(\tilde{y}, x, e) = f_X(x) f_{\varepsilon_1}(T(\tilde{Y}, X) - e) f_{\varepsilon_Y}(e) \tag{E.14}$$

$$= j(x, \tilde{y}) k(e, T(\tilde{Y}, X)), \tag{E.15}$$

for some functions $j, k$. Note that this means that, when conditioned on a constant $T(\tilde{Y}, X)$, the joint distribution $g_{\tilde{Y}, X, \varepsilon_Y}$ can be factorized into the product of two distributions; one involving $X, \tilde{Y}$, and the other involving $\varepsilon_Y, T(\tilde{Y}, X)$.

It follows that $T(\tilde{Y}, X)$ renders $\varepsilon_Y$ conditionally independent of $\tilde{Y}, X$:

$$\varepsilon_Y \perp\!\!\!\perp \tilde{Y}, X | T(\tilde{Y}, X). \tag{E.16}$$

Now, this implies that $T(\tilde{Y}, X)$ is sufficient for estimating the conditional expectation:

$$\mathbb{E}[\varepsilon_Y | \tilde{Y}, X] = \mathbb{E}[\varepsilon_Y | \tilde{Y}, X, T(\tilde{Y}, X)] = \mathbb{E}[\varepsilon_Y | T(\tilde{Y}, X)]. \tag{E.17}$$

Finally, note that

$$T(\tilde{Y}, X) = \tilde{Y} - f(X) \tag{E.18}$$

$$= \varepsilon_Y + f(X) + \varepsilon_1 - f(X) \tag{E.19}$$

$$= \varepsilon_Y + \varepsilon_1. \tag{E.20}$$

$$\implies \mathbb{E}[\varepsilon_Y | \tilde{Y}, X] = \mathbb{E}[\varepsilon_Y | \varepsilon_Y + \varepsilon_1]. \tag{E.21}$$

As $X$ is jointly independent of both $\varepsilon_Y, \varepsilon_1$ by assumption, it follows that $\mathbb{E}[\varepsilon_Y | T(\tilde{Y}, X)] \perp\!\!\!\perp X$.

Therefore, we conclude that $\mathbb{E}[\varepsilon_Y | \tilde{Y}, X] \perp\!\!\!\perp X$, and thus Eq E.10 holds.

### E.12.2 Anticausal Direction

If $f$ is linear, then by Lemma 4.1 we have $\mathbb{E}[\varepsilon_X | \tilde{X}, Y] \not\perp\!\!\!\perp Y$.

Suppose $f$ is nonlinear.

Let $h(\tilde{X}, Y) := \mathbb{E}[\varepsilon_X | \tilde{X}, Y]$. This proof will proceed in the following steps.

1. We will characterize $h(\tilde{X}, Y)$, showing that it must be a non-trivial function of $Y$ and $\tilde{X}$.

2. We will outline all possible cases (and subcases), in which $h(\tilde{X}, Y)$ is a non-trivial function of $Y$.

3. For each case (and associated subcases) we will show that $h(\tilde{X}, Y)$ is a function of a noise term dependent on $Y$.

4. We conclude that, as $h(\tilde{X}, Y)$ is always a function of noise dependent on $Y$, $h(\tilde{X}, Y) \not\perp\!\!\!\perp Y$.

Note that $h(\tilde{X}, Y)$ can be decomposed as $h(\tilde{X}, Y) = A_1(\tilde{X}) + A_2(Y) + A_3(\tilde{X}, Y)$, where $A_3$ contains only (linear or nonlinear) interaction between $\tilde{X}$ and $Y$, while $A_1, A_2$ are univariate.

In the graphical model (see Figure 4), there is an active path $Y \to \varepsilon_X$ when conditioning on the collider $\tilde{X}$. Due to d-separation rules [61], it follows that $\varepsilon_X \not\perp\!\!\!\perp Y | \tilde{X}$. Note that this implies that, under regularity conditions assumed above, the conditional distribution $P(\varepsilon_X | B, C) \neq P(\varepsilon_X | C)$ on a set of positive measure. This implies that $\mathbb{E}[\varepsilon_X | \tilde{X}, Y] \neq \mathbb{E}[\varepsilon_X | \tilde{X}]$, and therefore $h(\tilde{X}, Y)$ is a non-trivial function of $Y$. Therefore, at least one of $A_2, A_3$ is non-trivial. Similarly, as $\varepsilon_X \not\perp\!\!\!\perp \tilde{X} | Y$, at least one of $A_1, A_3$ must be non-trivial.

We will now walk through the following 3 cases: 1) that $A_3$ is non-trivial, 2) that $A_3$ is trivial and $A_2$ is non-trivial and $f$ is invertible, and 3) that $A_3$ is trivial and $A_2$ is non-trivial and $f$ is non-invertible. In each case we will show that $h(\tilde{X}, Y)$ is a function of a noise term dependent on $Y$. Case 2 will have 4 subcases.

Suppose $A_3$ is non-trivial. This implies that there exist interaction terms between $\tilde{X}, Y$. Suppose for contradiction that $h(\tilde{X}, Y)$ is not dependent on a noise term dependent on $Y$. Then, it implies that $A_1, A_2$ somehow cancel out all $Y$ terms in $A_3$. This leads to a contradiction, since the space of additive functions—i.e., those expressible as a linear combination of univariate functions of each variable—cannot represent interaction terms, which require non-additive combinations such as products of variables. Therefore, $h(\tilde{X}, Y)$ is a non-trivial function of $Y$, making it a non-trivial function of noise term $\varepsilon_1, \varepsilon_1 \not\perp\!\!\!\perp Y$.

Suppose $A_3$ is trivial. Then, $A_1$ and $A_2$ must be non-trivial.

Suppose $f$ is invertible. There are 4 possible subcases. In each case, $X$ is modelled by a different function of $Y$, and different noise. We list each case explicitly:

1. $X = f_1(Y) + e_1, Y \perp\!\!\!\perp e_1$

2. $X = f_2(Y, e_2), Y \perp\!\!\!\perp e_2$

3. $X = f_3(Y, e_3), Y \not\perp\!\!\!\perp e_3$

4. $X = f_4(Y) + e_4, Y \not\perp\!\!\!\perp e_4$

We note that, in Case 2 and 3 functions $f_2$ and $f_3$ induce nonlinear interactions between their inputs $Y, e_2$ and $e_3$.

Note that Case 1 cannot occur, as it violates Assumption 2.2 by allowing for the existence of a backwards model $X \to Y$ with additive noise.

Let's assume Case 2. Then,

$$h(\tilde{X}, Y) = A_1(f_2(Y, e_2) + \varepsilon_X) + A_2(Y).$$

As $f_2$ induces nonlinear interaction between $Y$ and $e_2$, where $e_2 \perp\!\!\!\perp Y, e_2 \perp\!\!\!\perp \varepsilon_X$, the collection of terms in $A_1$ containing $Y, e_2$ cannot be equal to a univariate function of $Y$. Therefore, the residual $r = A_1(f_2(Y, e_2) + \varepsilon_X) - A_2(Y)$ must be dependent on both $Y$ and $e_2$.

Let's assume Case 3. Then

$$h(\tilde{X}, Y) = A_1(f_3(Y, e_3) + \varepsilon_X) + A_2(Y).$$

Note that as $f_3$ induces nonlinear interaction between $Y$ and $e_3$, $e_3$ needs to be a function of $\varepsilon_1$ - if it would be only a function of $Y$, that would imply a deterministic relationship between $X$ and $Y$, which contradicts our setup. Due to the nonlinear dependence between $Y$ and $e_3$ in $f_3$, the $e_3$ term can not be canceled out by the univariate function $A_2(Y)$. Therefore, $h(\tilde{X}, Y)$ must contain the noise term $e_3, e_3 \not\perp\!\!\!\perp Y$.

Let's assume Case 4. Then

$$h(\tilde{X}, Y) = A_1(f_4(Y) + e_4 + \varepsilon_X) + A_2(Y) \tag{E.22}$$

Similar to the argument in Case 3, although $e_4 \not\perp\!\!\!\perp Y$, $e_4$ cannot solely be a function of $Y$ - this would again imply a deterministic relationship between $X, Y$. Therefore, $e_4$ must also be a function of $\varepsilon_1$. Then, the residual noise $r = A_1(f_4(Y) + e_4 + \varepsilon_X) + A_2(Y) \neq 0$, and $r \not\perp\!\!\!\perp Y$.

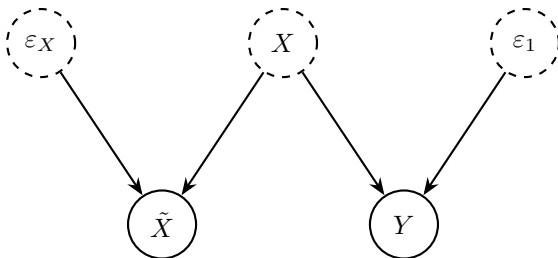

Figure 4: Dependence in the Anticausal Direction: $\tilde{X} = X + \varepsilon_X$ and $Y = f(X) + \varepsilon_1$. All root nodes are drawn from independent noise.

Therefore, $h(\tilde{X}, Y)$ is always a function of a noise term dependent on $Y$.

Suppose $f$ is non-invertible. Then there exists no backwards model where $X$ can be written as a function of $Y$ and noise. Therefore, $A_1$ cannot be written as a function of $Y$, noise and $\varepsilon_X$. Then, the residual $r = A_1 + A_2$ must contain $Y$, which means that $h(\tilde{X}, Y)$ contains $Y$ (which contains $\varepsilon_1$).

We have shown, either that $h$ always contains $\varepsilon_1$, or that $h$ contains a different term dependent on $Y$. Taken together our results imply that $Y \not\perp h(\tilde{X}, Y) \implies \mathbb{E}[\varepsilon_X|, \tilde{X}, Y] \not\perp Y$.

$\square$

### E.13 Challenges in Extending Theorem 4.2

The major challenge in extending Theorem 4.2 to the irreducible case, i.e., when ANM-UM does not reduce to ANM due to the presence of a nonlinear mediator, lies in the increased difficulty in analyzing the properties of the estimator $\mathbb{E}[\varepsilon_Y|\tilde{Y}, X]$. This difficulty arises when $X$ becomes non-additively related to the latent noise as a result of nonlinear mediators.

Note, our procedure rests on the intuition that noise predictions from a conditional diffusion model will be less dependent on the condition when the condition is the cause, rather than the effect.

When the ANM-UM reduces to ANM, we can actually show that *the noise prediction is entirely independent of the condition* when the condition is the cause. We can write $Y$ as a (potentially nonlinear) function of $X$ plus noise, i.e. $Y = f(X) + \varepsilon_1$. This clean additive separation between the noise $\varepsilon_1$ and $f(X)$ allows us to show that the predicted noise $\hat{\varepsilon} = \mathbb{E}[\varepsilon_Y|\tilde{Y}, X] = \mathbb{E}[\varepsilon_Y|\tilde{Y}, X, \tilde{Y} - f(X)] = \mathbb{E}[\varepsilon_Y|\tilde{Y}, X, \varepsilon_Y + \varepsilon_1] = \mathbb{E}[\varepsilon_Y|\varepsilon_Y + \varepsilon_1]$. This reformulation of $\hat{\varepsilon}$ into a term that is clearly independent of $X$ is the backbone of our proof of Theorem 4.2.

However, when the ANM-UM does not reduce to ANM, i.e. because of a nonlinear mediator, $Y$ can only be written as a function of $X$ and noise (Lemma 2.3): $Y = f(X) + g(\varepsilon) + h(X, \varepsilon)$, where $X, \varepsilon$ are nonlinearly combined in $h$. Here, its not possible to cleanly project $X$ out of $\tilde{Y}$, i.e., there does not exist a function $k(\cdot)$ such that $k(\tilde{Y}, X)$ is independent of $X$. Therefore, to extend Theorem 4.2 to this setting, we would need to show that $\mathbb{E}[\varepsilon_Y|\tilde{Y}, X]$ has a smaller dependence on $X$, when compared to the dependence between $\mathbb{E}[\varepsilon_X|\tilde{X}, Y]$ and $Y$. Analyzing the strength of the dependence would probably require a very different set of proof techniques, which are beyond the scope of this paper. One potential first step to bridge the gap between the reducible and irreducible settings is to identify conditions under which the interaction term $h(X, \varepsilon)$ becomes sufficiently 'weak', such that $\mathbb{E}[\varepsilon_Y|\tilde{Y}, X]$ becomes 'close to independent'. This will probably involve some investigation into the interaction by functional forms, noise distributions and the resulting mutual information between $X$ and the interaction term $h(X, \varepsilon)$.

# F  Details on Bivariate Denoising Diffusion (BiDD)

## F.1  Adapting Diffusion for Causal Discovery

BiDD works by training a diffusion model in both direction and selecting the direction with the lower noise prediction. We will now describe the details of training the diffusion model, estimating the mutual information, and deciding on causal direction. We describe the full procedure of BiDD in Algorithm 1. We formalized all three components as subroutines 1, 2 and 3.

---

**Algorithm 1** BiDD: Causal-direction discovery via conditional diffusion

---

**Require:** Training set $\mathcal{D}_{\text{Train}}$, Test set $\mathcal{D}_{\text{Test}}$, epochs $E$, timesteps $T$, mutual information estimator MI, noise schedule $\{\bar{\alpha}_t\}_{t \in [T]}$
**Ensure:** Causal direction $A \rightarrow B$ or $B \rightarrow A$
1: $\varepsilon_{A|B} \leftarrow$ TRAINCONDITIONALDIFFUSION($\mathcal{D}_{\text{Train}}, E, T, \{\bar{\alpha}_t\}_{t \in [T]}$)
2: $\{\text{MI}_{A,t}\}_{t \in [T]} \leftarrow$ ESTIMATEMUTUALINFORMATION($\mathcal{D}_{\text{Test}}, E, T, \text{MI}, \{\bar{\alpha}_t\}_{t \in [T]}, \varepsilon_{A|B}$)
3: $\varepsilon_{B|A} \leftarrow$ TRAINCONDITIONALDIFFUSION(swap($\mathcal{D}_{\text{Train}}$), $E, T, \{\bar{\alpha}_t\}_{t \in [T]}$)
4: $\{\text{MI}_{B,t}\}_{t \in [T]} \leftarrow$ ESTIMATEMUTUALINFORMATION(swap($\mathcal{D}_{\text{Test}}$), $E, T, \text{MI}, \{\bar{\alpha}_t\}_{t \in [T]}, \varepsilon_{B|A}$)
5: **return** COMPAREMUTUALINFORMATION($\{\text{MI}_{A,t}\}_{t \in [T]}, \{\text{MI}_{B,t}\}_{t \in [T]}$)

---

### F.1.1  Training Conditional Diffusion Model

Subroutine 1 adapts the denoising-diffusion training loop of Ho et al. [15, Alg. 1] to our conditional setting. In our training, we perform a single, full-dataset update per epoch.

---

**Subroutine 1** Train diffusion model

---

**Require:** Training set $\mathcal{D}_{\text{Train}} = \{(A^{(i)}, B^{(i)})\}_{i \in [n]}$, epochs $E$, total timesteps $T$, noise schedule $\{\bar{\alpha}_t\}_{t \in [T]}$
**Ensure:** Trained model $\varepsilon_\theta$
1: **for** $e = 1, \ldots, E$ **do**
2:    Independently sample timesteps $t^{(i)} \sim \mathcal{U}\{1, \ldots, T\}$ for all $i = 1, \ldots, n$
3:    Draw noises $\varepsilon^{(i)} \sim \mathcal{N}(0, 1)$ for all $i$
4:    Construct noised inputs

$$\tilde{A}_t^{(i)} = \sqrt{\bar{\alpha}_{t^{(i)}}}\, A^{(i)} + \sqrt{1 - \bar{\alpha}_{t^{(i)}}}\, \varepsilon^{(i)} \quad \text{for } i = 1, \ldots, n$$

5:    Compute loss

$$\mathcal{L} = \frac{1}{n} \sum_{i=1}^{n} \left\| \varepsilon^{(i)} - \varepsilon_\theta\big(\tilde{A}_t^{(i)}, B^{(i)}, t^{(i)}\big) \right\|^2$$

6:    Take gradient step on $\theta$
7: **end for**
8: **return** $\varepsilon_\theta$

---

### F.1.2  Mutual Information Estimation

After training the diffusion model, we use Subroutine 2 to estimate the mutual information between the condition and the predicted noise.

Note that in order to improve the finite sample performance of the mutual information estimators, the test set input to Subroutine 2 can be oversampled. This will lead to datapoints with identical conditions $B$, but differently noised version $\tilde{A}$. We use an oversampling factor $k = 10$ in our experiments.

### F.1.3  Deciding causal direction

After obtaining $\{MI_{A,t}\}_{i=1}^{T}$ (denoising $\tilde{A}$) and $\{MI_{B,t}\}_{i=1}^{T}$ (denoising $\tilde{B}$) for both directions, we decide the direction of the causal dependence.

---

**Subroutine 2** Estimate mutual information

---

**Require:** Test set $\mathcal{D} = \{A^{(i)}, B^{(i)}\}_{i \in [n]}$, epochs $E$, timesteps $T$, mutual information estimator MI, noise schedule $\{\bar{\alpha}_t\}_{t \in [T]}$, trained diffusion model $\varepsilon_\theta$
**Ensure:** Series of MI estimations $\{\text{MI}_t\}_{t \in [T]}$
    **for** $t = 1, \ldots, T$ **do**
        Draw random noise $\varepsilon^{(i)} \sim N(0, 1)$ for all $i$.
        Generate $\tilde{A}_t^{(i)}$ from test set as $\tilde{A}_t^{(i)} = \sqrt{\bar{\alpha}_t}\, A^{(i)} + \sqrt{1 - \bar{\alpha}_t}\, \varepsilon^{(i)}$
        Predict $\hat{\varepsilon}_A^{(i)} = \varepsilon_\theta(\tilde{A}_t^{(i)}, B^{(i)}, t^{(i)})$
        Calculate $MI_t = MI(\{\hat{\varepsilon}_A^{(i)}\}_{i=1}^n, \{B^{(i)}\}_{i=1}^n)$
    **end for**
    **return** $\{\text{MI}_t\}_{t \in [T]}$

---

**Voting rule** For the voting rule, we conclude $B \to A$ if and only if $MI_{A,t} < MI_{B,t}$ for the majority of timesteps. We use this rule in the main body of the paper. We formalize it in Subroutine 3.

---

**Subroutine 3** Compare mutual information

---

**Require:** MI sequences $\{\text{MI}_{A,t}\}_{t \in [T]}$ and $\{\text{MI}_{B,t}\}_{t \in [T]}$
**Ensure:** Chosen causal direction ($A \to B$ or $B \to A$)
  1: $v \leftarrow \sum_{i=1}^T \mathbf{1}\{MI_{A,i} < MI_{B,i}\}$
  2: **if** $v > T/2$ **then**
  3:      **return** $B \to A$ k
  4: **else**
  5:      **return** $A \to B$
  6: **end if**

---

**Mean rule** For the mean rule, we conclude $B \to A$ if and only if $\frac{1}{T}\sum_i^T \text{MI}_{A,i} < \frac{1}{T}\sum_i^T \text{MI}_{B,i}$. We present results on this decision rule in Appendix H.

## F.2 Implementation details

Models were implemented in `PyTorch`[45]. Training employed the `AdamW` optimizer. We used a Hilbert–Schmidt Independence Criterion (HSIC) implementation in `PyTorch`. For the alternative mutual-information estimator evaluated in Appendix H, we used the `NPEET` package.

### F.2.1 Mutual-information estimation

Dependence between the predicted noise and the conditioning variable is quantified with HSIC [10], which we treat as a surrogate for mutual information. HSIC is computed with a Gaussian kernel whose bandwidth is selected via the median pairwise-distance heuristic. The robustness of our results to the choice of dependence measure is examined in Appendix H.

### F.2.2 Training hyperparameters

In our diffusion setup, we use $T = 256$ timesteps, scheduling $\beta$ linearly from $\beta_{\min} = 0.0001$ to $\beta_{\max} = 0.02$. For stochastic gradient descent, we use the AdamW optimizer with cosine annealing. We set an initial learning rate of $0.0001$ and decay to $0.00001$. We train our model for a total of 4000 epochs. For BiDD$_{\text{Test}}$, we used $80\%$ of the data for training and evaluated performance on the remaining $20\%$. For BiDD$_{\text{Total}}$, we trained on $80\%$ of the data and evaluated on the entire dataset.

### F.2.3 Model Architecture

The diffusion model for BiDD uses an MLP architecture to predict $\varepsilon(\tilde{A}_t, B, t)$. Each of the three inputs is first fed through a dedicated input projection: a $1 \to 512$ linear layer for $\tilde{A}_t$, a small conditioning network for $B$, and a sinusoidal time embedding followed by two SiLU-activated linear layers for $t$. The resulting $(512 + 4 + 512) = 1028$-d feature vector is concatenated and processed

| Stage | Operation / Activation | Output shape |
|---|---|---|
| *Input projections* | | |
| $\tilde{A}_t$ proj. | Linear $(1{\rightarrow}512)$ | $(B, 512)$ |
| $B$ proj. | Linear $(1{\rightarrow}16)$+ReLU | $(B, 16)$ |
| | Linear $(16{\rightarrow}32)$+ReLU | $(B, 32)$ |
| | Linear $(32{\rightarrow}4)$ | $(B, 4)$ |
| $t$ embedding | Sinusoidal $(1{\rightarrow}16)$ | $(B, 16)$ |
| | Linear $(16{\rightarrow}512)$+SiLU | $(B, 512)$ |
| | Linear $(512{\rightarrow}512)$+SiLU | $(B, 512)$ |
| CONCAT | — | $(B, 1028)$ |
| *Residual MLP blocks* (repeat twice) | | |
| Hidden | Linear $(1028{\rightarrow}2056)$+SiLU | $(B, 2056)$ |
| | Linear $(2056{\rightarrow}1028)$ | $(B, 1028)$ |
| Residual add | $x \leftarrow x + \text{block}(x)$ | $(B, 1028)$ |
| *Output projection* | | |
| Output proj. | Linear $(1028{\rightarrow}512)$+SiLU | $(B, 512)$ |
| | Linear $(512{\rightarrow}1)$ | $(B, 1)$ |

Table 5: Layer specification for the BiDD denoising model predicting $\varepsilon(\tilde{A}_t, B, t)$. $B$ is batch size. SiLU is the Sigmoid-weighted Linear Unit, ReLU is the Rectified Linear Unit.

by two residual MLP blocks, each expanding to two times width and returning to the original size. A final projection $(1028{\rightarrow}512{\rightarrow}1)$ produces the noise estimate $\varepsilon_\theta(\tilde{A}_t, B, t)$. We document the exact layer setup in Table 5. Overall, the model contains **9,260,893** learnable parameters.

# G  Experimental details

## G.1  Evaluation Data

### G.1.1  Synthetic Data

To generate the synthetic data, we used different link functions in combination with different noise types.

**Details for the link functions**

$$\text{Quadratic:} \quad f(x) = \left(x\right)^2 + \varepsilon,$$

$$\text{Tanh:} \quad f(x) = \tanh\!\left(x + o\right) + \varepsilon, \qquad\qquad o \sim \mathcal{U}(-1, 1),$$

$$\text{Linear:} \quad f(x) = ax + o + \varepsilon, \qquad\qquad a \sim \mathcal{U}(-5, 5), o \sim \mathcal{U}(-3, 3),$$

$$\text{Neural network:} \quad f(x) = \tanh\!\left(x\, \mathbf{w}_{\text{in}}^\top + \mathbf{1}\, \mathbf{b}_{\text{h}}^\top\right) \mathbf{w}_{\text{out}} + \varepsilon,$$

where the weight vectors lie in $\mathbb{R}^h$ and are sampled i.i.d. from

$$\mathbf{w}_{\text{in}}, \ \mathbf{b}_{\text{h}}, \ \mathbf{w}_{\text{out}} \ \sim \ \mathcal{U}(-5, 5).$$

The parameters $o, a, \mathbf{w}_{\text{in}}, \ \mathbf{b}_{\text{h}}, \ \mathbf{w}_{\text{out}}$ are randomly drawn for each mediator and run.

**Noise types**   We evaluated BiDD on two different noise types: uniform and Gaussian. For noise inside the mediators ($\varepsilon_1$ till $\varepsilon_T$ in Figure 1), we used noise with mean 0 and variance .5. For noise generating X and Y ($\varepsilon_0$ and $\varepsilon_{T+1}$ in Figure 1), we used noise with mean 0 and variance 1.

**Data generating process**   We used the same noise type for generating the cause and noise in the process for our experiments to generate $X$ and $Y$. For each mediator, we redrew the random

parameters for the link functions. More specifically, to synthesise a single cause–effect observation $(X, Y)$ with $T$ latent mediators, we begin by drawing the cause $X \sim \mathcal{D}(0, 1)$, where $\mathcal{D}(\mu, \sigma^2)$ denotes the chosen base noise family (e.g. $\mathcal{N}$ or $\mathcal{U}$). Setting $Z_0 = X$, we traverse a chain of $T$ unobserved mediators. For each index $j = 1, \ldots, T$ we independently *(i)* sample a noise term $\varepsilon_j \sim \mathcal{D}(0, 0.5)$ and *(ii)* select a link function $g_j$ at random. The mediator is then evaluated as

$$Z_j = g_j(Z_{j-1}) + \varepsilon_j.$$

After the final mediator we draw $\varepsilon_{T+1} \sim \mathcal{D}(0, 1)$ and an additional link function $f_{T+1}$, generating the effect variable by

$$Y = f_{T+1}(Z_T) + \varepsilon_{T+1}.$$

Finally, both $X$ and $Y$ are centered and rescaled to unit variance. Repeating this procedure independently yields an i.i.d. dataset that follows an ANM-UM with $T$ unobserved mediators.

### G.1.2 Real-world Data

We evaluated the performance on the first 99 pairs of the Tübingen dataset [42], loaded from the `causal-discovery-toolbox` package. For each pair, we randomly subsampled 3000 data points in each run for faster execution. We calculated the accuracy as the simple average over all 99 pairs.

### G.2 Implementation Details

We coded our experiments using python `3.11.11` with `PyTorch 2.5.1` [45], and ran the experiments on AWS `g4dn.xlarge` ec2 instances. We provide our public repo at `https://github.com/dommeier/bidd`.

### G.3 Baseline Implementation

We imported CANM from the code provided in Cai et al. [6]. We did not use the described method to find the correct number of mediators, but instead called the method with the correct number of mediators. We trained the VAE for 2000 epochs.

DirectLiNGAM and RESIT were imported from the `lingam` package. CAM, SCORE and NoGAM were imported from the `dodiscover` package. CAM-UV was imported from the `lingam` package. Adascore was imported using the `causal-score-matching` package. Var-Sort was implemented using `NumPy`. DagmaL was imported from the `dagma` package. PNL was implemented following the logic in the `causal-learn` library, but with slight modifications in order to execute model training on GPU for faster execution.

We list the hyperparameters we used in Table 6.

| Method | Hyperparameters |
|---|---|
| DirectLiNGAM | None |
| CAM | `prune=False` |
| RESIT | `RandomForestRegressor(max_depth=4)` |
| SCORE | `prune=False` |
| NoGAM | `n_crossval=2`, `prune=False` |
| var_sort | None |
| entropy_knn | `k=100`, `base=2` |
| PNL | None |
| AdaScore | `alpha_orientation=0.05 alpha_confounded_leaf=0.05` `alpha_separations=0.05` |
| DagmaL | `lambda1=0.05`, `T=4`, `mu_init=1` `s=[1, 0.9, 0.8, 0.7]`, `mu_factor=0.1` |

Table 6: Hyperparameters used for each method.

## G.4 Runtime

We provide the mean runtime for 80 independent runs across all mechanisms (linear, tanh, neural network, quadratic) and noises (Gaussian, uniform), measured as wall-clock time. The runtime experiments were conducted on AWS `g4dn.xlarge` ec2 instances with GPU support, with no parallelization.

All methods that do not rely on stochastic-gradient training finish in under 2 seconds. Among the baselines, PNL is an order of magnitude slower because it trains a small neural network.

The scenario with unobserved mediators is inherently more complex than fully observed settings tackled by the baselines, which justifies a longer runtime. When compared with the only other method that explicitly addresses unobserved mediation, CANM, BiDD has comparable runtimes.

For PNL, BiDD, and CANM the wall-clock time is governed mainly by (i) the size of the training set, (ii) the complexity of the model, and (iii) the number of training epochs. We didn't exhaustively tune model size, epoch count, or code efficiency, so runtimes could be reduced with further optimisation.

| Method | Runtime (s) |
|---|---|
| BiDD | 172.0 |
| CANM | 145.5 |
| Adascore | 1.25 |
| NoGAM | 0.23 |
| SCORE | 0.28 |
| DagmaL | 0.85 |
| CAM | 0.15 |
| PNL | 35.80 |
| RESIT | 0.43 |
| DLiNGAM | 0.01 |
| Var-Sort | 0.00 |

Table 7: Average runtimes of methods for $n = 1000$ in seconds.

## G.5 Asset information

All external code we import is open-source under permissive licences: `lingam`, `dodiscover`, `causal-learn`, and `NPEET` are MIT; `causal-score-matching` is MIT-0; `dagma` is Apache-2.0.

We use the Tübingen cause–effect dataset curated by Mooij et al. [40]. Several of its variable pairs originate from datasets released by Kelly et al. [24] in the UCI Machine Learning Repository.

# H Ablations

## H.1 Mutual information estimate

### H.1.1 Setup

To test the sensitivy of BiDD to the choice of mutual information estimator, we test it's performance under two different mutual information estimators. We test three different sets of hyperparameters for each of them.

For HSIC, in our main experiments, we heuristically pick the width of the Gaussian kernel as the median in the distance metric. As an ablation, we scale this width by .5 and 2.

As an alternative estimator, we use the `NPEET` package, which implements the non-parametric estimator by Kraskov et al. [26], using k-nearest neighbors approach. We vary the number of neighbors $k = 3, 5, 10$.

We use the same experimental setup as in our previous synthetic experiments, setting $n = 1000$, and using a test split of $20\%$.

| Method
Noise | Linear
Unif. | Neural Net
Gauss. | 
Unif. | Quadratic
Gauss. | 
Unif. | Tanh
Gauss. | 
Unif. | Average |
|---|---|---|---|---|---|---|---|---|
| **Full data, voting** | | | | | | | | |
| HSIC(1) | 0.80 | **0.95** | 0.90 | **1.00** | **1.00** | 0.85 | 0.80 | 0.85 |
| HSIC(.5) | 0.85 | **0.95** | 0.85 | **1.00** | 0.80 | **0.90** | 0.80 | 0.83 |
| HSIC(2) | 0.80 | 0.90 | 0.90 | **1.00** | **1.00** | 0.85 | 0.80 | **0.86** |
| NPEET(3) | 0.85 | **0.95** | 0.85 | **1.00** | 0.70 | 0.80 | **0.95** | 0.81 |
| NPEET(5) | **0.90** | **0.95** | 0.85 | **1.00** | 0.70 | 0.75 | **0.95** | 0.81 |
| NPEET(10) | **0.90** | 0.90 | 0.85 | **1.00** | 0.60 | 0.80 | **0.95** | 0.82 |
| **Full data, mean** | | | | | | | | |
| HSIC(1) | 0.80 | **0.95** | 0.95 | **1.00** | **1.00** | 0.80 | 0.70 | 0.84 |
| HSIC(.5) | 0.85 | 0.90 | 0.90 | 0.80 | 0.75 | **0.90** | 0.70 | 0.79 |
| HSIC(2) | 0.70 | **0.95** | **1.00** | **1.00** | **1.00** | 0.70 | 0.65 | 0.83 |
| NPEET(3) | **0.90** | **0.95** | 0.85 | **1.00** | 0.65 | 0.70 | **0.95** | 0.81 |
| NPEET(5) | **0.90** | **0.95** | 0.85 | **1.00** | 0.55 | 0.75 | **0.95** | 0.81 |
| NPEET(10) | 0.85 | **0.95** | 0.80 | **1.00** | 0.45 | 0.75 | **0.95** | 0.79 |
| **Test data, voting** | | | | | | | | |
| HSIC(1) | 0.80 | 0.85 | 0.80 | **1.00** | 0.95 | 0.75 | 0.70 | 0.79 |
| HSIC(.5) | 0.75 | 0.90 | 0.85 | **1.00** | 0.75 | 0.75 | 0.75 | 0.76 |
| HSIC(2) | 0.65 | 0.90 | 0.90 | **1.00** | **1.00** | 0.80 | 0.65 | 0.80 |
| NPEET(3) | 0.75 | 0.85 | 0.80 | **1.00** | 0.70 | 0.70 | 0.90 | 0.79 |
| NPEET(5) | 0.70 | 0.85 | 0.85 | **1.00** | 0.80 | 0.60 | 0.90 | 0.78 |
| NPEET(10) | 0.75 | 0.90 | 0.85 | **1.00** | 0.85 | 0.70 | 0.90 | 0.81 |
| **Test data, mean** | | | | | | | | |
| HSIC(1) | 0.75 | 0.85 | 0.90 | **1.00** | 0.95 | 0.65 | 0.65 | 0.79 |
| HSIC(.5) | 0.75 | **0.95** | 0.90 | 0.95 | 0.60 | 0.85 | 0.70 | 0.77 |
| HSIC(2) | 0.75 | 0.85 | **1.00** | **1.00** | **1.00** | 0.60 | 0.65 | 0.81 |
| NPEET(3) | 0.75 | 0.80 | 0.85 | **1.00** | 0.45 | 0.75 | 0.75 | 0.74 |
| NPEET(5) | 0.80 | 0.80 | 0.80 | **1.00** | 0.40 | 0.70 | 0.75 | 0.73 |
| NPEET(10) | 0.80 | **0.95** | 0.90 | **1.00** | 0.55 | 0.70 | **0.95** | 0.80 |

Table 8: Accuracy of various MI estimators across transform–noise combinations. Best scores per column are in **bold**, second–best are underlined. Accuracy is averaged over 20 runs per mechanism. HSIC($s$): Factor $s$ applied to kernel width. NPEET($k$): Number of neighbors $k$

For every run we report two accuracy criteria: (i) *voting*, which counts a direction as correct if the majority of timesteps agree; (ii) a *mean*, which averages the MI over all timesteps and picks the lower-dependence direction (see Section F.1.3).

## H.1.2 Results

Complete results appear in Table 8.

**Overall robustness.** Every configuration achieves at least 76% mean accuracy, rising to 79% or better when the full data set is used for estimating mutual information.

**Estimator–mechanism interactions.** HSIC dominates on the *quadratic + uniform-noise* mechanism, whereas NPEET is best on *tanh + uniform*. These preferences are consistent across both voting and mean rules.

**Voting vs. mean.** Discrepancies between the two decision rules widen on the test split. For example, NPEET with $k=3$ misses the *quadratic + uniform* case under the mean rule but is working good under voting. Conversely, HSIC occasionally loses $5\% - 10\%$ on *tanh + normal* when switching from voting to mean.

**Hyperparameters for estimators.** Within each estimator family, hyper-parameter choices have second-order impact: HSIC's wider bandwidth ($\times 2$) and NPEET's larger neighbourhoods ($k=5$ or 10) bring modest, but consistent, improvements.

| Method Noise | Linear Unif. | Neural Net Gauss. | Neural Net Unif. | Quadratic Gauss. | Quadratic Unif. | Tanh Gauss. | Tanh Unif. |
|---|---|---|---|---|---|---|---|
| **With conditioning:** | | | | | | | |
| $\text{BiDD}_{\text{Total}}$ | **0.83** | 0.87 | 0.97 | **1.00** | **1.00** | **0.80** | **0.83** |
| $\text{BiDD}_{\text{Test}}$ | 0.80 | 0.87 | **1.00** | **1.00** | **1.00** | 0.63 | 0.77 |
| **No conditioning:** | | | | | | | |
| $\text{BiDD}_{\text{Total}}$ | 0.15 | **0.90** | 0.75 | 0.30 | 0.35 | 0.60 | 0.05 |
| $\text{BiDD}_{\text{Test}}$ | 0.15 | 0.85 | 0.80 | 0.05 | 0.55 | 0.40 | 0.10 |

Table 9: Conditioning is necessary for identification across different mechanisms: Results for same setup as Table 2, but with modified training objective without conditioning. We report the mean accuracy over 20 runs.

## H.2 Conditioning vs Non-Conditioning

### H.2.1 Setup

To test how conditioning impacts the performance of our methods, we provide an ablation study where we train on an unconditional loss. That means that the diffusion model does not have access to the conditioning variable $B$ anymore, and needs to predict the noise $\varepsilon$ only from $\tilde{A}$ and $t$. We replace the original loss function:

$$L_{\text{CDM}} = \mathbb{E}_{A,B,\varepsilon,t} \left[ \|\varepsilon - \varepsilon_\theta(\tilde{A}_t, B, t)\|^2 \right], \quad t \sim \text{Unif}(\{1,\ldots,T\})$$

with the unconditional one:

$$L_{\text{DM}} = \mathbb{E}_{A,\varepsilon,t} \left[ \|\varepsilon - \varepsilon_\theta(\tilde{A}_t, t)\|^2 \right], \quad t \sim \text{Unif}(\{1,\ldots,T\}),$$

and keep the setup and training identical.

### H.2.2 Results

**Conditioning is important** Table 9 shows the importance of conditioning. While accuracy in the *neural-network* setting remains similar, it deteriorates across all other mechanisms, and the model often selects wrong directions for both *linear + uniform* and *tanh + uniform*. These results indicate that conditioning the diffusion model is an important part for reliable causal discovery in our framework.

## H.3 Two mediators

| Method Noise | Linear Unif. | Neural Net Gauss. | Neural Net Unif. | Quadratic[3] Gauss. | Quadratic Unif. | Tanh Gauss.[4] | Tanh Unif. |
|---|---|---|---|---|---|---|---|
| $\text{BiDD}_{\text{Total}}$ | 0.65 | **0.80** | **0.95** | 0.70 | 0.30 | 0.45 | 0.85 |
| $\text{BiDD}_{\text{Test}}$ | 0.70 | 0.75 | **0.95** | 0.70 | **0.60** | 0.45 | 0.85 |
| RESIT | **0.90** | 0.50 | 0.60 | 0.00 | 0.00 | **0.60** | **0.90** |
| PNL | 0.55 | 0.55 | 0.55 | **0.75** | 0.40 | 0.30 | 0.50 |

Table 10: Accuracy of methods across different transformation-noise combinations, with *two mediators* and n=1000.

---

[3]Results for the quadratic link function with a high number of mediators tend to become unreliable due to numerical instability from repeated quadratic operations, as reported in previous work [13]

[4]Results for the tanh link function and Gaussian noise can become unreliable if sample size is not high enough, because the Tanh function is close to linear across -1 to 1, where most of the mass of a Gaussian distribution is, rendering the empirical distribution close to the unidentifiable linear Gaussian case.

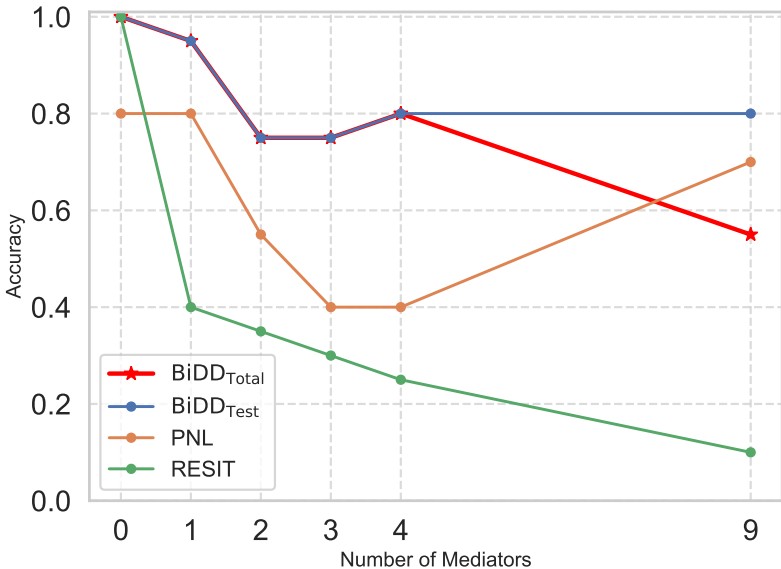

Figure 5: Accuracy for Neural Network + Gaussian setting with varying number of mediators

### H.3.1 Setup

All baselines, except PNL and RESIT, fail in at least two settings, achieving accuracy of at most 50%. This renders them unreliable in scenarios where only the ANM-UM assumption holds. Of the three remaining methods, all use residual independence as the decision criterion, so their difference in performance probably depends on how much the underlying model violates ANM or PNL. While they might perform well in settings with a low number of mediators, they might fail with an increased number of mediators.

To analyze how RESIT and PNL perform when their assumptions are violated, we re-ran the synthetic experiments using the same setup as in Table 2, but with two mediators.

### H.3.2 Results

While the average performance of all methods declines when increasing the number of mediators from one to two mediators (e.g., from 0.90 to 0.67 for BiDD$_{\text{Total}}$, 0.87 to 0.71 for BiDD$_{\text{Test}}$, 0.76 to 0.51 for PNL, and 0.88 to 0.50 for RESIT), the drop is particularly pronounced for PNL and RESIT in settings with neural net link functions: Both BiDD variants consistently achieve at least 75% accuracy across noise types, whereas the best baseline score does not exceed 60%.

We explain this difference in performance with a DGP which is not following the assumptions from PNL or RESIT anymore, while still following the ANM-UM, which BiDD can exploit.

### H.4 More mediators in non-invertible Gaussian settings

### H.4.1 Setup

We extend the experiments for Figure 2a to the Neural Network + Normal setting, to investigate how a non-invertible mechanism and noise combination behaves under varying number of mediators.

### H.4.2 Results

We report the accuracy of 20 runs across different methods in Figure 5. For no mediator, all methods perform well. When increasing the number of mediators, both PNL and RESIT exhibit a sharp decline in performance for one mediator, and keep declining for more mediators. Both BiDD versions are more stable and decline slower.

