# OpenReview forum: "When Additive Noise Meets Unobserved Mediators: Bivariate Denoising Diffusion for Causal Discovery"
_NeurIPS.cc/2025/Conference — NeurIPS 2025 poster_

### Official Review · Reviewer_wazL · 2025-07-03

**Clarity:** 3
**Significance:** 3
**Originality:** 3
**Rating:** 4
**Confidence:** 4

**Summary:**

This paper discusses the problem of causal direction identification under the addictive noise model with unmeasured mediator (ANM-UM). The authors first analyze the equivalent condition for ANM-UM to reduce to common ANM. Then, they analyze how existing methods based on ANM can fail under unmeasured mediator. Next, under the linear setup, they develop an identification rule of the causal direction via denoising. Experiments on synthetic and real-world datasets are conducted for verification.

**Questions:**

**1.** Can the authors explain the major challenge when extending Thm. 4.2 to the irreducible case? Is there any potential idea for the extension?

**2.** If BiDD works under the irreducible case (as conjectured in Sec. 4.2) and the baselines do not work under ANM-UM (as shown in Sec. 3), why their performance seems to be comparable?

**Ethical Concerns:**

["NO or VERY MINOR ethics concerns only"]

**Final Justification:**

The author's response has partly addressed my concern. So I keep my positive score.

**Limitations:**

Yes

**Quality:**

2

**Strengths And Weaknesses:**

**Strength**

**1**. This paper establishes the necessary and sufficienct condition for the ANM-UM to reduce to the ANM.

**2**. The authors conducts exhausive analyses on why existing methods fail under unmeasured mediation.

**3**. They conduct both synthetic and real-world experiments for validation.

**Weakness**

**1**. The consistency of the decision rule is shown only for the case where ANM-UM reduces to ANM. For the irreducible case, e.g., nonlinear causal relationship, there is no theoretical intuition why this method can work.

Indeed, the authors consider empirical verifications in Sec. 5. But the improvement of BiDD over baseline methods seems to be insignificant. I not sure whether this result can support the conjecture that BiDD also works under the irreducible case.

---

> ### Author Rebuttal · Authors · 2025-07-31
>
> We thank the reviewer for highlighting our establishment of the "necessary and sufficient condition for the ANM-UM to reduce to the ANM", our "exhaustive analysis" of failure modes in existing methods, and our use of "synthetic and real-world experiments for validation". We also appreciate the reviewer's insightful questions, which has helped us improve the clarity and conceptual strength of our work. In what follows, **we address all points raised**.
>
> **Weaknesses and Questions (combined)**
>
> **(W1+Q1)** *Extending Thm 4.2 to Irreducible Case*
>
> We first note that Theorem 4.2 is non-trivial and, to our knowledge, represents the first theoretical result on using denoising diffusion for bivariate causal discovery—not merely to estimate score function Jacobians, as in prior work [1], but to reveal asymmetries arising from the independence structure inherent to ANMs.
>
> The major challenge in extending Theorem 4.2 to the irreducible case, i.e., when ANM-UM does not reduce to ANM due to the presence of a nonlinear mediator, lies in the increased difficulty in analyzing the properties of the estimator $E[\varepsilon_Y|\tilde{Y},X]$. This difficulty arises when $X$ becomes non-additively related to the latent noise as a result of nonlinear mediators.
>
> Note, our procedure rests on the intuition that noise predictions from a conditional diffusion model will be less dependent on the condition when the condition is the cause, rather than the effect.
>
> When the ANM-UM reduces to ANM, we can actually show that _the noise prediction is entirely independent of the condition_ when the condition is the cause. We can write $Y$ as a (potentially nonlinear) function of $X$ plus noise, i.e. $Y = f(X)+\varepsilon_1$. This clean additive separation between the noise $\varepsilon_1$ and $f(X)$ allows us to show that the predicted noise $\hat\varepsilon=E[\varepsilon_Y|\tilde{Y},X]= E[\varepsilon_Y|\tilde{Y},X, \tilde{Y}-f(X)]=E[\varepsilon_Y|\tilde{Y},X, \varepsilon_Y+\varepsilon_1]= E[\varepsilon_Y|\varepsilon_Y+\varepsilon_1]$. This reformulation of $\hat\varepsilon$ into a term that is clearly independent of $X$ is the backbone of our proof of Theorem 4.2.
>
> However, when the ANM-UM does not reduce to ANM, i.e. because of a nonlinear mediator, $Y$ can only be written as a function of $X$ and noise (Proposition 2.3): $Y = f(X) + g(\varepsilon) + h(X,\varepsilon)$, where $X,\varepsilon$ are nonlinearly combined in $h$. Here, its not possible to cleanly project $X$ out of $\tilde{Y}$, i.e., there does not exist a function $k(\cdot)$ such that $k(\tilde{Y},X)$ is independent of $X$. Therefore, to extend Theorem 4.3 to this setting, we would need to show that $E[\varepsilon_Y|\tilde{Y},X]$ has a smaller dependence on $X$, when compared to the dependence between $E[\varepsilon_X|\tilde{X},Y]$ and $Y$. Analyzing the strength of the dependence would probably require a very different set of proof techniques, which are beyond the scope of this paper. One potential first step to bridge the gap between the reducible and irreducible settings is to identify conditions under which the interaction term $h(X,\varepsilon)$ becomes sufficiently 'weak', such that $E[\varepsilon_Y|\tilde{Y},X]$ becomes 'close to independent'. This will probably involve some investigation into the interaction by functional forms, noise distributions and the resulting mutual information between $X$ and the interaction term $h(X,\varepsilon)$ .
>
> **(W2+Q2)** *Comparable performance of the baselines*
>
> For a method to be effective, it should work across all mechanisms that follow the ANM-UM, not only selected ones. In our synthetic experiments, we therefore tried to incorporate different types of noise (Normal, Uniform) and different invertible (Linear, Tanh) and non-invertible (Neural Net, Quadratic) link functions.
>
> All baselines, except PNL and RESIT, fail in at least two settings, achieving accuracy of at most 50\%. This renders them unreliable in scenarios where only the ANM-UM assumption holds.
>
> *Performance of RESIT, PNL, BiDD for two mediators*
> Of these three methods, all use residual indepenced, so their difference in performance probably depends on how much the underlying model violates ANM or PNL.
> While they might perform well in settings with a low number of mediators, they might fail with an increased number of mediators.
>
> To analyze how RESIT and PNL perform when their assumptions are violated, we re-ran the synthetic experiments using the same setup as in Table 2, but with two mediators:
> |                      |     Linear ,  Unif.   |     Neural Net ,  Gauss.   |     Neural Net ,  Unif.   |     Quadratic$^A$ ,  Gauss.   |     Quadratic$^A$ ,  Unif.   |     Tanh ,  Gauss.$^B$   |     Tanh ,  Unif.   |     Tanh ,  Unif.   |
> |---------------------|------------------------|-------------------------------|--------------------------------|--------------------------|---------------------------|---------------------|----------------------|----------------------|
> | $BiDD_\textrm{Total}$           |                    0.65 |                           0.80 |                            0.95 |                      0.70 |                       0.30 |                 0.45 |                  0.85 |                  1.00 |
> | $BiDD_\textrm{Test}$ |                    0.70 |                           0.75 |                            0.95 |                      0.70 |                       0.60 |                 0.45 |                  0.85 |                  0.80 |
> | RESIT        |                    0.90 |                           0.50 |                            0.60 |                      0.00 |                       0.00 |                 0.60 |                  0.90 |                  0.90 |
> | PNL                  |                    0.55 |                           0.55 |                            0.55 |                      0.75 |                       0.40 |                 0.30 |                  0.50 |                  0.50 |
>
> [$A$] Results for the quadratic link function with high number of mediators tend to become unreliable, because the repeated application of the quadratic operation tends to lead to numerical instability, as reported in previous work [2].
> [$B$] Results for the Tanh link function + Gaussian noise can become unreliable if sample size is not high enough, because the Tanh function is close to linear across -1 to 1, where most of the mass of a Gaussian dist. is, rendering the empirical distribution close to the unidentifiable linear Gaussian case.
>
> While the average performance of all methods declines when increasing the number of mediators (e.g., from 0.86 to 0.68 for $BiDD_\textrm{Total}$, 0.82 to 0.72 for $BiDD_\textrm{Test}$, 0.75 to 0.52 for PNL, and 0.83 to 0.54 for RESIT), the drop is particularly pronounced for PNL and RESIT in settings with neural net link functions. Both BiDD variants consistently achieve at least 75\% accuracy across noise types, whereas the best baseline score does not exceed 60\%.
>
>
> **References**
>
> ${[}1{]}$ Sanchez et al. "Diffusion models for causal discovery via topological ordering". ICLR, 2023.
>
> ${[}2{]}$ Hiremath et al. "Hybrid top-down global causal discovery with local search for linear and nonlinear additive noise models." Advances in Neural Information Processing Systems, 2024.

---

> > ### Comment · Reviewer_wazL · 2025-08-03
> >
> > I thank the authors for responding. I will keep my positive score.

---

> > > ### Author Response · Authors · 2025-08-03
> > >
> > > We thank the Reviewer for their consideration.

---

### Official Review · Reviewer_UfQZ · 2025-07-05

**Clarity:** 3
**Significance:** 2
**Originality:** 3
**Rating:** 4
**Confidence:** 2

**Summary:**

This paper proposes Bivariate Denoising Diffusion (BiDD), a method for identifying causal direction in the presence of unobserved mediators. BiDD leverages diffusion models, hypothesizing that noise predictions are less dependent on the cause than the effect. Empirical evaluations demonstrate BiDD’s superiority over baselines on synthetic and real-world data.

**Questions:**

- What properties of diffusion (e.g., noise-perturbation process) make them more effective in handling unobserved mediators? Can you offer a theoretical analysis to it?
- What is the identifiability when nonlinear mediator exists.

**Ethical Concerns:**

["NO or VERY MINOR ethics concerns only"]

**Final Justification:**

Thank you for the authors’ response. Most of my concerns have been addressed, and I will raise my score accordingly. I recommend that the key differences from [1] be well discussed in the final version.

**Limitations:**

See Weaknesses.

**Paper Formatting Concerns:**

NAN

**Quality:**

2

**Strengths And Weaknesses:**

Strengths:
- BiDD adapts diffusion models to bivariate causal discovery.
- Extensive experiments validate BiDD’s performance.

Weaknesses:
- First, the paper does not adequatly explain why diffusion models are theoretically better suited for this task compared to alternatives like VAEs.
- Decision Rule 1 appears incorrect, as it ignores the edge case where \( MI(\epsilon_Y, X) = MI(\epsilon_X, Y) \).
- Theorem 4.2 only addresses identifiability when no nonlinear mediator exists. The paper should further provide guarantees for the more general case where nonlinear mediators are present.

---

> ### Author Rebuttal · Authors · 2025-07-31
>
> We thank the reviewer for recognizing the "superiority over baselines" and appreciating the "extensive experiments". We also value the reviewer's thoughtful questions, which have helped us improve both the clarity and conceptual rigor of our work. In the following, **we respond to all points raised**.
>
> **Weaknesses and Questions (combined)**
>
> **(W1+Q1)** *Diffusion models vs. VAE*
>
> While both VAEs and diffusion models are generative frameworks, how this paper and the CANM paper used them was both theoretically different and practically different.
>
> *Theoretical analysis:*
> CANM [1], based on a VAE, minimizes a reconstruction loss combined with a regularization term on the latent space. This does not align directly with the assumptions of the additive noise model with unobserved variables (ANM-UM). Other methods relying on MSE-Minimization therefore need to include other assumptions for theoretical guarantees (see Section 3 --> Failure Modes -> MSE Minimization) As discussed in Section 3, such approaches may fail even under ANM-UM, as proven in Lemma 3.4 (*MSE-Minimization Fails*)
>
> In contrast, BiDD utilizes a diffusion model trained via mean squared error (MSE), but evaluated through an independence test between the predicted noise and the conditioning variable. This explicitly encodes asymmetry in the data-generating process, as implied by the ANM-UM assumption (independence of residuals in one direction only). As a result, our approach yields a formal theoretical guarantee for causal identifiability in the ANM setting—something the VAE-based method is unable to provide.
>
> Another difference between VAE and Diffusion based methods, beyond how they are specifically implemented in CANM and BiDD, is that VAE guarantees are likely to depend on whether the dimensionality of the latent space, i.e. the number of unobserved mediators assumed by a VAE method, actually matches the true number of unobserved mediators in the causal DGP.  In contrast, denoising Diffusion does not require an estimate of an unknowable parameter for its procedure. Therefore, apriori, for tasks where aspects of the underlying causal structure (such as the number of mediators) are both important and hard to ascertain, we propose that it is theoretically justified to prefer Diffusion over VAEs.
>
> *Empirical performance:*
> In practice, VAEs are known to be sensitive to hyperparameters and prone to instability. Their training objective includes two components—the KL divergence of the latent representation and the reconstruction loss—and the relative strength of these terms is highly dependent on optimization details. Avoiding posterior collapse often requires extensive tuning or auxiliary tricks.
>
> Even with an improved training schedule designed to mitigate posterior collapse (see Appendix E.9), we observed poor performance of the VAE baseline in synthetic experiments with invertible link functions (Tanh + Linear) and uniform noise. In contrast, our diffusion-based method trains robustly across a range of configurations and performs consistently well across all evaluated settings.
>
> In summary, our choice of diffusion modeling is both theoretically motivated and practically effective, offering advantages over VAE-based alternatives in the context of causal discovery with unobserved mediators.
>
>
>
> **(W2)** *Edge case for decision rule 1*
>
> We thank the reviewer for pointing out that edge case.
> We updated our decision rule so that we do not decide when the edge case of equality occurs:
> > **Decision rule 1** (Bivariate Denoising Diffusion (BiDD))
> Let $\hat\varepsilon_Y=\varepsilon_{Y,\theta}(\widetilde{Y},X)$, $\hat\varepsilon_X=\epsilon_{X,\theta}(\widetilde{X},Y)$ be the predictions of the noise added to $Y,X$, respectively.
> Given a mutual information estimator $MI(\cdot)$,
> if $MI(\hat\varepsilon_Y, X) < MI(\hat\varepsilon_X, Y)$, conclude that $X$ causes $Y$. If $MI(\hat\varepsilon_Y, X) > MI(\hat\varepsilon_X, Y)$, conclude that $Y$ causes $X$. Else, do not decide.
>
> We will apply the same update to the other decision rules presented in the paper to ensure consistency.
>
> **(W3 + Q2)** *Theoretical consistency in the nonlinear mediator case*
>
> Our theoretical analysis assumes a linear mediator, as this preserves a simple Additive Noise Model (ANM) structure even when the mediator is unobserved. In contrast, nonlinear mediators break this structure—making the relationship between cause and effect no longer additive. This loss of structure complicates the analysis, as it does not allow for a relatively straightforward decomposition anymore. However, we argue that our proof of consistency in ANM (Theorem 4.2) is an important first step, and an extension is nontrivial: for more context on these issues, we refer to our answer to Q1 of Reviewer wazL.
>
> **References**
>
> ${[}1{]}$ Cai et al. "Causal discovery with cascade nonlinear additive noise models." arXiv, 2019.

---

> > ### Comment · Reviewer_UfQZ · 2025-08-05
> > **Thank you for the authors’ response.**
> >
> > Thank you for the authors’ response. Most of my concerns have been addressed, and I will raise my score accordingly. I recommend that the key differences from [1] be well discussed in the final version.

---

> > > ### Author Response · Authors · 2025-08-05
> > >
> > > We thank the Reviewer for their follow-up and are happy to clarify any remaining concerns that may not have been fully addressed in our initial response.
> > >
> > >
> > > We **confirm** that we will include an expanded discussion in Section 3.2 (Hidden Mediator Method – CANM) to differentiate our approach from [1]. Specifically:
> > >
> > > (1) *Diffusion vs VAE*
> > >
> > > We will highlight the general advantage of the diffusion model framework over the VAE framework for generative modeling in this scenario, which is greater robustness to hyperparameters such as the latent dimensionality.
> > >
> > > (2) *CANM([1]) vs BiDD (this work)*
> > >
> > > We will also clarify the key differences between [1] and this work:
> > > - [1] provides no theoretical guarantees. In contrast, our method offers theoretical guarantees under the ANM assumption.
> > > - [1] evaluates causal direction using lower reconstruction loss (lower MSE), which does not directly align with the assumptions of the additive noise model with unobserved mediators (ANM-UM). In contrast, our method leverages the asymmetry implied by the ANM-UM assumption to identify causal direction.

---

### Official Review · Reviewer_mAkY · 2025-07-08

**Clarity:** 3
**Significance:** 3
**Originality:** 2
**Rating:** 4
**Confidence:** 3

**Summary:**

This paper aims at inferring causal direction between two variables when unobserved mediators are present with nonlinear relations with the output , where additive noise model (ANM) assumptions may not hold. The author provided a through discussion why existing methods failed, and then proposed a Bivariate Denoising diffusion that is provable to learn the causal relation under ANM. Experiments on synthetic and real-world data are conducted.

**Questions:**

See the weakness above.

**Ethical Concerns:**

["NO or VERY MINOR ethics concerns only"]

**Final Justification:**

The authors have addressed most of my concerns. I maintain my positive score.

**Limitations:**

Yes.

**Quality:**

2

**Strengths And Weaknesses:**

Strengths.

1. The paper considered an important scenario in bivariate causal discovery, and was overlooked by the literature.
2. The authors rigorously analyze when and why existing methods fail under hidden mediators, and provide consistency guarantees for BiDD under standard ANM.
3. BiDD outperforms existing methods on synthetic datasets across a range of mechanisms (linear, nonlinear, invertible/non-invertible).

Weakness.

1. The theoretical consistency of BiDD is only proved in the standard ANM setting, and only conjectured in more realistic settings with nonlinear mediators.
2. Training two conditional denoising diffusion models and estimating mutual information is computationally intensive, which may limit scalability to large datasets.
3. Assumption 2.2 is difficult to decipher. It is difficult to comprehend which constraints on f1,...,fN and noise to make the assumption holds.

---

> ### Author Rebuttal · Authors · 2025-07-31
>
> We thank the reviewer for recognizing that our work addresses an "important scenario", provides a "rigorous analysis of when and why existing methods fail", offers consistency guarantees for ANM, and demonstrates superior performance on synthetic datasets.
>
> We appreciate the reviewer’s constructive feedback, which has helped us enhance both the clarity and conceptual depth of our work. In the following, we address all points raised:
>
> **Weaknesses**
>
> **(W1)** *Theoretical consistency*
>
> We agree that a theoretical consistency guarantee for BiDD in the general case of ANM-UM, beyond our consistency result in the ANM, would significantly strengthen our contribution. However, we argue that our proof of consistency in ANM (Theorem 4.2) is an important first step, and an extension is nontrivial due to the presence of unobserved mediators, which introduces fundamental identifiability challenges. For more context on these issues, we refer to our answer to Q1 of Reviewer wazL.
>
> **(W2)** *Scalability*
>
> We can decompose the scalability of our approach into two aspects:
> a) the ability to handle larger datasets or higher-dimensional input/output spaces, and
> b) the runtime of the method when used as a subroutine, a common scenario for bivariate causal discovery, e.g. in global discovery methods.
>
>
> - **(W2a)** *Ability to handle large datasets*
>
>   - While diffusion models are computationally intensive, they are known to scale well to large datasets. For high-dimensional data, prior work has proposed latent diffusion models [1], which operate in a compressed latent space to significantly reduce memory and compute requirements while maintaining generation quality.
>
> - **(W2b)** *Runtime of BiDD*
>
>   - The scenario with unobserved mediators is inherently more complex than fully observed settings, which justifies a longer runtime. Nonetheless, when compared with the only other method that explicitly addresses this case, the CANM, BiDD is competitive:
>
> | Method | Runtime (s) |
> |--------|-------------|
> | BiDD   | 172.0       |
> | CANM   | 145.5       |
>
>
> While the primary focus of this work is on providing a theoretically grounded and consistent method, we agree that improving runtime is a valuable direction for future research. Since BiDD involves both training and generation, advances in accelerating diffusion models apply directly. Prior work has proposed techniques to make training more efficient [2], to speed up generation [3], and to fine-tune pre-trained models for downstream tasks rather than training from scratch [4], that all could be applicable here. We will include this as a potential future direction at the end of the paper.
>
> **(W3)** *Clarity of Assumption 2.2*
>
> Our identification results rely on the assumption that the noise injected into each unobserved mediator is mutually independent, which means that we can express $Y$ as
> $$
>     Y = F(X, \varepsilon_1,\ldots,\varepsilon_{T})+\varepsilon_{T+1}
> $$
> where $X,\varepsilon_1,\ldots,\varepsilon_{T+1}$ are all mutually independent, and $F=f_{T+1}(f_T(\ldots)+\varepsilon_{T})$.
> For the ANM-UM to be identifiable, it is essential that there does not exists a Backward ANM-UM, a model that satisfies the ANM-UM assumptions in the reverse direction ($Y \rightarrow X$)
> This would require the existence of functions $G$ and noise terms $\hat\varepsilon_1,\ldots,\hat\varepsilon_{T+1}$ such that $X$ can be decomposed as a function of $Y$ and independent noise terms, i.e.,:
> \begin{align}
>     X = G(Y, \hat\varepsilon_1,\ldots,\hat\varepsilon_T)+\hat\varepsilon_{T+1}
> \end{align}
> where $G=g_{T+1}(g_T(\ldots)+\hat{\varepsilon_{T}})$
> and, additionally, $Y,\hat\varepsilon_1,\ldots,\hat\varepsilon_{T+1}$ are mutually independent.
> For a formal discussion of the conditions under which backward ANMs can arise, we refer the reviewer to Appendix D.2.
>
> To improve clarity and provide additional intuition for the Backward ANM-UM concept, we will revise the paragraph preceding Assumption 2.2 in the main text as follows:
>
> > Prior work (Theorem 1, [Cai et al., 6]) shows that certain joint distributions over $(X, Y)$ admit ANM-UM representations in both directions, $X \dashrightarrow Y$ and $Y \dashrightarrow X$, rendering the causal direction unidentifiable without further assumptions. We refer to such distributions as Backward ANM-UM, as the structural assumptions of ANM-UM hold in both the causal and anticausal directions. These distributions arise only under pathological conditions, such as when all functions are linear and the noises are Gaussian.
>
>
> **References**
>
> ${[}1{]}$ Rombach et al. "High-resolution image synthesis with latent diffusion models." CVPR, 2022.
>
> ${[}2{]}$ Choi et al. "Perception prioritized training of diffusion models." CVPR, 2022.
>
> ${[}3{]}$ Song et al. "Denoising diffusion implicit models." arXiv, 2022.
>
> ${[}4{]}$ Watson et al. "De novo design of protein structure and function with RFdiffusion." Nature 620, 2023.

---

### Official Review · Reviewer_UBVq · 2025-07-10

**Clarity:** 3
**Significance:** 3
**Originality:** 3
**Rating:** 5
**Confidence:** 2

**Summary:**

The proposed method in this paper offers a causal direction identification method while addressing the hidden mediator case. The authors show precisely where existing work does not perform well and why. The proposed method uses denoising diffusion to obtain a new conditional independence test.  Finally, they demonstrated empirical performance on both synthetic and real-world datasets.

**Questions:**

Below I share my questions.

* Why can we not consider all $Z_i$ as a single variable $Z$ and then apply existing methods?
* The authors mentioned (line 202) that “inclusion may also introduce dependence between the predicted noise and the unnoised variable”. Why does this happen?
* Line 271: Is it possible that the inferred direction keeps flipping at different timestamps $t$? What does this behavior indicate? Does the method eventually converge to a specific direction?
* (Line 297) For which setup was the causal direction the hardest to infer?
* How many mediators can the proposed method accommodate before performance degrades?

**Ethical Concerns:**

["NO or VERY MINOR ethics concerns only"]

**Final Justification:**

The authors have addressed my concerns and added additional experiments. Thus, I keep my positive score.

**Limitations:**

yes

**Quality:**

3

**Strengths And Weaknesses:**

Below I share my comments.

### Strength:
* The paper is written in a clear and well-organized manner. The proposed method appears to be theoretically sound.
* I appreciate the authors' discussion on the failure modes of prior works in different scenarios. The analysis of failure cases is helpful for understanding the motivation and contributions of the proposed method.

### Weakness:
* Line 198: Typo. I think it should be “noising/forward process”.
* Line 200: In $(\epsilon_Y - f_{\epsilon_Y} (\tilde{Y}, X))^2$, why is it $X$ and not $\tilde{X}$, given that noises were added to both $\tilde{X}$ and $\tilde{Y}$? Are the noising processes in Eq $4.1$ and $4.2$ applied to $X$ and $Y$ at the same time, or are they applied individually while keeping the other variable fixed? The authors should clarify this point.
* The authors mentioned that they choose the direction with lower dependence (mutual information). Shouldn’t there be no dependence in the anti-causal direction? What does it mean when both directions have high dependence (but unequal), or both have low dependence (but unequal)? What does the gap between the two dependence values imply? These questions should be addressed in the paper.

---

> ### Author Rebuttal · Authors · 2025-07-31
>
> We thank the reviewer for highlighting several strengths of our submission, including the discussion of failure modes in existing work described as “helpful for understanding the motivation and contributions of the proposed method”, as well as the “clear and well-organized” presentation and the “sound” methodology.
>
> We also appreciate the reviewer’s thoughtful questions, which have helped improve both the clarity and depth of our work. Below, we address each of the points raised in detail.
>
>
> **Weaknesses**
>
> **(W1)** *Typo*
>
> We thank the reviewer for pointing out the typo. We address it with our update to the paragraph in (W2).
>
>
>
>
> **(W2)** *Noising process*
>
> We formulate two noising/denoising processes. One for each direction $X \rightarrow Y$ and $Y \rightarrow X$.
> In each of the processes, we keep one variable fixed, while the other is noised. To clarify this setup, we will update the paragraph in Section 4.2. containing line 200:
>
> > To show the asymmetry in this setup, we formulate two noising processes: one where we keep $X$ fixed and noise $Y$, and one where we keep $Y$ fixed and noise $X$. In the noising processes, we inject independent Gaussian noise into the noised variable, obtaining the noised terms
> \begin{align}
>     \widetilde{X} = X + \varepsilon_X~~~ \text{and}~~~
>     \widetilde{Y} = Y + \varepsilon_Y, ~~~ \varepsilon_X,
>     \varepsilon_Y, \sim \mathcal{N}(0,1).
> \end{align}
> Now, in the denoising processes, we aim to find the best estimators
> $f_{\varepsilon_X}, f_{\varepsilon_Y}$ such that MSE losses
> \begin{align}
>     (\varepsilon_Y-f_{\varepsilon_Y}(\widetilde{Y},X))^2 ~~~\text{and}~~~ (\varepsilon_X-f_{\varepsilon_X}(\widetilde{X},Y))^2
> \end{align}
> are minimized.
>
> **(W3)** *Dependence in the anti-causal direction*
>
> As $X$ and $Y$ are functionally related, conditioning on either variable when predicting the added noise $\hat{\varepsilon}$ can induce dependence between the predicted noise and the conditioning variable, potentially in both directions. In our analysis of the ANM sub-case, we show that in the causal direction, the predicted noise remains independent of the conditioning variable, whereas in the anticausal direction, statistical dependence is introduced (Theorem 4.2).
>
> Under the ANM-UM, $Y$ can be modelled as a series of transformations of $X$ where at each step an independent error is injected additively. Intuitively, we hypothesize that the additive noise is easier to decouple from $X$ when denoising $Y$ while conditioning on $X$, thereby rendering the estimate of the noise added to $Y$ ($\hat{\varepsilon_Y}$) less dependent on $X$. In contrast, in the anticausal direction, $X$ can only be modelled as a series of transformations of $Y$ where the error is _not additively injected_; intuitively, this would implying that denoising $X$ while conditioning on $Y$ would lead to a predicted noise that is more dependent on $Y$.
>
> Empirically, we observe a similar pattern empirically in ANM-UM: the amount of dependence differs between directions, with the causal direction consistently yielding the smaller dependence. The magnitude of the estimated dependence is influenced by several factors (link function, noise dist, etc.).  Therefore, we focus on the gap between the dependencies in the two directions as the signal for inferring causal direction.
>
> We have included a summary of this discussion in the paragraph before Section 4.1 on page 5.
>
>
>
> **Questions**
>
> **(Q1)** *Multiple $Z_i$ or only one $Z_i$*
>
> The core challenge in causal discovery with unobserved mediators lies not in their number, but in the fact that all $Z_i$ are unobserved. Existing methods cannot be applied in this setting, as they rely on access to observed mediators. In contrast, our method is explicitly designed to handle entirely unobserved mediation.
>
> We choose to model the process with multiple latent variables, instead of collapsing all $Z_i$ into one $Z$, to preserve the independent additive noise structure. If we used only one latent $Z$, then when there is nonlinear mediator $Z_i$ in the causal chain between $X$ and $Y$, then $Y$ would be a non-additive function of $Z$ and noise. This breaks the underlying ANM structure we exploit.
>
>
> **(Q2)** *How inclusion introduces dependence*
>
> To clarify why adding this conditional information might introduce dependence, we will add the following paragraph in Section 4.1 after `However, this inclusion may also introduce dependence between the predicted noise and the unnoised variable`:
>
> > Consider the case where $X$ is the cause and $Y$ is the effect. Under the ANM-UM assumption, this implies that $Y$ is a function of $X$ and independent noise terms. Let $(X_i, Y_i)$ be a fixed sample from the dataset. Suppose we add noise to $Y_i$ to obtain a noised version $\tilde{Y}_i = Y_i + \varepsilon_Y$. When predicting $\varepsilon_Y$ from $\tilde{Y}_i$ and $X_i$, the estimate $\hat{\varepsilon}_Y$ may depend on $X_i$, since $X_i$ contains information about the true value of $Y_i$. This induces statistical dependence between $X$ and the predicted noise $\hat{\varepsilon}_Y$. The same argument applies in the anticausal direction. However, the dependence is not symmetric, as the functional relationship between $X$ and $Y$ is not symmetric. This asymmetry in conditional dependence provides a useful signal for causal identification and motivates our use of a conditional diffusion model.
>
> **(Q3)** *Timsteps*
>
> We would like to clarify that the timestep $t$ in our model does not correspond to an iterative sampling process with varying outputs. Rather, it represents a fixed point in the denoising trajectory of the learned diffusion model.
>
> At each timestep $t$, we generate predictions of the added noise in both directions, $\hat\varepsilon_{X | Y}$ and $\hat\varepsilon_{Y | X}$. For each direction, we provide the model with three inputs: (1) the conditioning variable (either $X$ or $Y$), (2) the noised variable (either $\tilde{Y}$ or $\tilde{X}$), and (3) the timestep $t$.
> We then compute the dependence between the predicted noise and the conditioning variable, yielding a dependence estimate for each direction at each timestep.
>
> Empirically, we found that comparing the dependence estimates across timesteps (i.e., using a majority vote) outperformed simply averaging them over time. For a detailed comparison of how this choice affects BiDD’s performance, we refer the reader to Appendix F.1, where we evaluate both the voting and averaging strategies empirically.
>
>
> **(Q4)** *Differences in setups*
>
> We investigated several factors that influence the difficulty of inferring the correct causal direction, including the choice of linking mechanism and noise type (Table 2), as well as the number of mediators and sample size (Figures 2a and 2b).
>
> We found that BiDD generally performs better when the link function is non-invertible and when the noise follows a uniform distribution (Table 2). As expected, increasing the sample size consistently improves performance (Figure 2b). The effect of the number of mediators is more nuanced, and depends strongly on the specific combination of linking function and noise type. We provide a detailed analysis of this interaction in our response to *(Q5)* below.
>
> **(Q5)** *Number of mediators*
>
> Generally, as the number of mediators increases, the dependence between $X$ and $Y$ weakens, and in the limit (under finite samples), they become independent. In such cases, our method will output each direction with approximately 50\% probability.
>
> The number of mediators that our method can accommodate depends on the specifics of the data-generating process. To determine the influence of different link functions and noise types, we re-ran the synthetic exps. using two mediators instead of one:
> |                      |  Linear, Uniform | Neural Net, Normal | Neural Net, Uniform | Quadratic$^A$, Normal | Quadratic$^A$, Uniform | Tanh, Normal $^B$| Tanh, Uniform |
> |-----------------------|----------------|--------------------|---------------------|-------------------|--------------------|--------------|---------------|
> | $BiDD_\textrm{Total}$  |       0.65       |         0.80         |         0.95          |        0.70         |         0.30         |      0.45      |       0.85      |
> | $BiDD_\textrm{Test}$   |       0.70       |         0.75         |         0.95          |        0.70         |         0.60         |      0.45      |       0.85      |
>
> [$A$] Results for the quadratic link function with high number of mediators tend to become unreliable, because the repeated application of the quadratic operation tends to lead to numerical instability, as reported in previous work [1]. [$B$] Results for the Tanh link function + Gaussian noise can become unreliable if sample size is not high enough, because the Tanh function is close to linear across -1 to 1, where most of the mass of a Gaussian dist. is, rendering the empirical distribution close to the unidentifiable linear Gaussian case.
>
> We observe that BiDD’s performance with two mediators remains stable in some settings (e.g., Tanh + Uniform), while it degrades significantly in others (e.g., Linear + Uniform). To investigate this behavior further, we extended our synthetic experiments by increasing the number of mediators for two cases: Tanh + Uniform and Neural Net + Normal. The results are summarized below:
> | Number of mediators     | 1   | 2    | 3    | 4    | 5    | 10   |
> |-------------------------|-----|------|------|------|------|------|
> | $BiDD_\textrm{Total}$ (Tanh + Uniform)          | 1.0 | 0.9  | 0.8  | 0.95 | 0.85 | 0.95 |
> | $BiDD_\textrm{Total}$ (Neural Net + Normal) | 1.0 | 0.95 | 0.75 | 0.75 | 0.8  | 0.55 |
>
> While performance remains relatively robust for Tanh + Uniform, the Neural Net + Normal setting exhibits a clear decline.
>
> We propose to add the results of both additional experiments to the paper.
>
>
> **References**
>
> ${[}1{]}$ Hiremath et al. "Hybrid top-down global causal discovery..." NeurIPS 2024.

---

> > ### Author Response · Authors · 2025-08-07
> >
> > Dear Reviewer UBVq,
> >
> > The Author-Reviewer discussion period is nearing the end. We hope our responses addressed your concerns. If you feel that they have, we would greatly appreciate your acknowledgement of our rebuttal. If you still have questions, please let us know! Thank you again for the detailed comments and questions, which have helped improve the quality of our work.

---

### Note · Authors · 2025-08-11

We thank all reviewers for their thoughtful feedback and for recognizing the strengths of our work, including its “clear and well-organized” presentation (UBVq), “sound methodology”, “rigorous” analysis of failure modes (mAkY), and use of “synthetic and real-world experiments” for validation (wazL). Reviewers uniformly acknowledged the significance of our work as the first method to target the ANM-UM setting, providing theoretical guarantees alongside robust empirical performance for handling unmeasured mediators.

The reviewers’ feedback offered valuable opportunities to improve the paper. As detailed in our individual rebuttal responses, we strengthened the manuscript through three key updates:

## 1. Differences to related work

We expanded our comparison between BiDD and CANM (Cai et al.), emphasizing both theoretical differences (BiDD offers guarantees under ANM; CANM does not) and methodological differences (dependence-based decision vs. MSE minimization).

This clearer positioning within the literature makes BiDD's contributions and novelty more explicit.

## 2. Additional experimental results

We added additional experiments to evaluate BiDD’s robustness as the number of unobserved mediators increases, and to compare its performance against baseline methods.

**Two-mediators setting**

We find that BiDD maintains consistent performance, while methods whose core assumptions are violated (e.g., RESIT, PNL) degrade significantly.

**Multiple-mediators setting**

BiDD remains robust in some mechanism/noise configurations (e.g., Tanh + Uniform), while performance naturally declines in others (e.g., Neural Net + Normal).

Evaluating BiDD across different mediator counts illustrates its empirical performance and shows that it outperforms baselines when their assumptions break.

## 3. Theoretical coverage for the ANM-UM case

Our theoretical guarantees currently apply only to the reducible ANM-UM case. The main challenge in extending the theory is the loss of the clean additive separation underlying Theorem 4.2. We outlined possible extensions as a promising direction for future work.

Clarifying our scope helps readers understand this choice and see potential next steps for extending BiDD’s theory to the ANM-UM case.

## Summary

These clarifications, together with the new results, address the reviewers’ key points and further strengthen both the theoretical framing and empirical evaluation of BiDD.

---

### Decision · Program_Chairs · 2025-09-17

**Decision:**

Accept (poster)

**Comment:**

This paper introduces Bivariate Denoising Diffusion (BiDD), a method for causal direction identification in the presence of unobserved mediators under the Additive Noise Model with Unmeasured Mediators (ANM-UM) setting.

Strengths:

- The authors articulates why existing methods fail under unmeasured mediation and motivate BiDD's use of denoising diffusion models for causal discovery

- The paper provides an analysis of failure modes in existing methods and establishes conditions for ANM-UM reducibility.

Weaknesses:

- theoretical guarantees only apply to the reducible ANM-UM case.

- proposed method is computationally intensive, potentially limiting applicability to large-scale problems.

- empirical improvements are sometimes marginal